**communications** engineering

# A miniature ultrasonic surgical device based on a flextensional configuration with a pre-stressed piezoelectric stack
Xuan Li [1,2], Dominic Jones[3], Pietro Valdastri [3] & Margaret Lucas [1] ✉

Ultrasonic bone scalpels are known to offer benefits of low cutting force, high precision, low microdamage around the cut site, and tissue selectivity in surgical procedures. However, all current commercial devices are too large to be integrated with the flexible endo-wrist of a surgical robot, and therefore, there is a significant gap for innovation in miniature devices. Ultrasonic bone scalpels in use in clinical settings are all based on a bolted Langevin transducer (BLT), which consists of a pre-stressed piezoceramic ring stack, two end masses, and a cutting blade. The BLT-based device must operate in resonance to achieve sufficient displacement amplitude at the surgical tip to cut through bone, and this dictates its size. Flextensional transducers have emerged as an alternative, but these transducers generally contain a low volume of piezoelectric driving material, and hence cannot excite the required displacement amplitude, and their reliance on adhesive bonds in their fabrication means they fail at the excitation levels required for a bone surgery device. We present a flextensional configuration that forms an ultrasonic surgical device, where the vibration-amplifying metal caps are excited by a pre-stressed piezoelectric stack. In vitro ultrasonic bone cutting tests facilitated with a Kuka robot are performed for a range of cutting speeds and penetration rates. The results demonstrate effective integration with a Kuka robot and that bone cutting can be achieved with an extremely low cutting force ( < 1 N) and high precision (the width of the bone cut presents under 6% deviation from the thickness of the blade). The flextensional device overcomes both the large size of conventional ultrasonic osteotomy devices and the displacement amplitude limitations of other miniaturisation approaches. Integration with an articulated robotic endo-wrist is enabled, establishing a foundation for low-force and high-precision ultrasonic bone cutting in minimally invasive, anatomically constrained robotic surgical environments.

The benefits of ultrasonic bone surgery devices are well established and include high precision, low cutting force, reduced trauma, less collateral damage, and sparing of other tissue structures[1–5]. Ultrasonic bone surgery devices operate at a low ultrasonic frequency, usually between 20 and 35 kHz, causing bone fragmentation due to high frequency impacting in the cut depth direction for scalpel-like blades or superposition of high frequency oscillations on the sawing action of the serrated blades. Ultrasonic bone surgery was originally invented for maxillofacial, periodontal, and endodontic surgeries[6,7], but applications have extended to many other surgical procedures, including skull base and spinal surgeries[3,8–11].

Conventional minimally invasive surgeries present practical limitations. For example, laparoscopic/endoscopic surgeries, through single or multiple incisions, use long and rigid surgical instruments that often suffer from the 'fulcrum' effect caused by the point of insertion into the body[12–15]. This fulcrum acts as a point of rotation, which inverts the surgeon's movements and can amplify hand tremor, making the instrument more difficult to use. Surgical robots have been introduced to clinical practice to mitigate these limitations, enhance dexterity, improve stability, and increase motion accuracy[12]. This technology is known as robotically assisted minimally invasive surgery (RAMIS), which originates from

[1]Centre for Medical & Industrial Ultrasonics, James Watt School of Engineering, University of Glasgow, Glasgow, UK. [2]Department of Mechanical Engineering, School of Engineering, University of Southampton, Southampton, UK. [3]STORM Lab UK, School of Electronic & Electrical Engineering, University of Leeds, Leeds, UK. ✉e-mail: Margaret.Lucas@glasgow.ac.uk

laparoscopic surgical procedures facilitated by the da Vinci surgical robot[16–19].

Incorporating ultrasonic surgical devices into RAMIS creates additional challenges. A long straight waveguide attached to the bolted Langevin transducer (BLT) is required to excite vibration of the distal surgical tip, but these can suffer from flexural vibrations, and hence heating[20,21]. Also, this configuration of the ultrasonic device is restricted in its degrees of freedom at the surgical site. The smaller, hand-held, ultrasonic devices cannot integrate with surgical robots; they do not fit through the trocar, which is typically 10 to 12 mm diameter[13–15,22–25], and they cannot connect to the robot arm. An alternative, much more dexterous method for delivering surgical instruments is through integration with the robot's endo-wrist, which can enable seven degrees-of-freedom. However, to connect with an endo-wrist and comply with the trocar size, a miniature ultrasonic surgical device is required.

Miniaturisation presents challenges due to the small volume of usable piezoelectric material, making the device less powerful than larger commercial ultrasonic surgical devices. Driving a miniature, but conventionally configured BLT at higher power is not a solution; the result includes detrimental thermal effects, significant loss in piezoelectric performance, and undesirable nonlinear behaviour. Additionally, because a shorter BLT is achieved by increasing the resonance frequency, a much lower vibrational displacement amplitude is excited in the cutting blade[4] and the performance needed for effective bone cutting cannot be achieved.

Alternative miniaturisation strategies for ultrasonic surgical devices, such as planar designs[26,27] and folded structures[28], have also been investigated. However, similar to the BLT configuration, reducing the length of planar devices raises the resonance frequency, limiting the achievable blade displacement for effective bone cutting. Folded structures may generate sufficient blade displacement at a resonance frequency suitable for bone cutting, but the associated stress concentrations at the fold corners significantly increase the risk of structural failure.

A flextensional transducer is another alternative configuration for device miniaturisation. This type of transducer is commonly used in low to medium ultrasonic frequencies and high-power underwater projectors, which radiate sound by the flexure of a metal shell excited by a piezoelectric plate operating in extensional vibration[29]. Lab-based prototypes of ultrasonic surgical devices based on flextensional configuration have been reported that demonstrate cutting of bone tissue[30–32]. However, the size is still too large for entry via a trocar (the largest diameter of a standard trocar for minimally invasive surgery is 15 mm[24]) due to the need to accommodate sufficient piezoelectric material to achieve the required displacement amplitude of the cutting blade.

A major issue with the configuration of the conventional classes of flextensional transducer is that they rely on a bonding agent (usually an epoxy resin for class IV (moonie), V (cymbal) and VI transducers[33]) to secure the metal shell to the piezoelectric material and this softens and fails at the high excitation levels required[32], limiting the devices to low displacement amplitude applications. Mechanical fixings along with an adhesive bond can enable higher displacement amplitudes to be reached[31,32], but the achievable displacement is still too low, and it does not solve the problem of the size of the device being too large for integration with a surgical robot endo-wrist.

Other classes of flextensional transducers (class I, II, III, and VII[33] barrel-stave transducers) employ a piezoelectric stack radially compressed by concave barrel-staves[34]. Although epoxy bonding is not essential for these designs, the radial stave geometry imposes a minimum practical diameter, while the requirement for a surrounding shell further limits their miniaturisation potential. In addition, these transducers are predominantly used in underwater applications and typically operate at resonance frequencies below 20 kHz[35].

Due to all the design constraints of these classes of flextensional transducer, a new design for a miniature ultrasonic surgical device is proposed that overcomes size, performance, and robotic integration limitations inherent to both the BLT and conventional flextensional transducer configurations. The design, which is based on a flextensional configuration, is presented in Fig. 1(a). The device is distinctive from the known classes of flextensional transducer[29,33], which employ piezoelectric plates or discs or bars as the ultrasonic vibration source, and then transform the small extensional motion of the driving element into flexural motion of the metal shell[31,32,36]. The novelty of this new device is that it employs a pre-stressed stack of piezoelectric rings connected to a single-piece metal frame, which functions as the mechanical amplifier and incorporates the surgical cutting blade. We demonstrate that prototypes based on this design are capable of sustaining high excitation levels for a long duration in bone cutting experiments, without showing signs of deterioration in their cutting performance. This is the first report of a miniature ultrasonic surgical device prototype that has been successfully integrated with a Kuka robot for hard tissue resection. The device is intended for minimally invasive surgical procedures such as neurospinal surgeries, where its compatibility with the articulated endo-wrist mechanism offers enhanced dexterity and access in confined anatomical spaces. The device performance in mock-ups of these specific surgeries will be investigated in future work.

## Results and discussion
### Miniature ultrasonic surgical device configuration and characterisation

The miniature ultrasonic surgical device was designed in CAD and modelled in finite element analysis (FEA), Fig. 1a. The device is comprised of a stack of lead zirconate titanate (PZT) piezoelectric rings pre-stressed in a single-piece metal frame by a threaded bar and two nuts. The metal frame has one working side, which incorporates the surgical blade, and one connecting side, which provides a connector to the Kuka robot (Fig. 1(b)). The metal frame, therefore, acts as a one-sided flextensional transducer, with an integrated serrated cutting blade. Design optimisation of the metal frame was carried out by varying key geometric parameters, thickness of the frame and the height of its apex, to evaluate the trade-offs between resonance frequency, structural stress, and displacement amplification. The geometry of the surgical blade, featuring serrations on one side and a sharp edge on the other side, enables investigation of different bone cutting mechanisms (sawing and chiselling).

Unlike the standard classes of flextensional transducer that are symmetrical and exhibit a symmetric mode of vibration (high deformation on both sides), the surgical device requires a non-vibrating connection to the Kuka robot. The geometry of the connection side of the device is therefore carefully designed to meet a number of requirements: (i) it must be configured to ensure that there is no vibration at the robot connection location, (ii) the mass and geometry must balance the overall vibrational mode such that the PZT stack deforms without any bending, (iii) it must not limit the vibrational amplification of the working flextensional side with the cutting blade, (iv) it must prevent distortion of the frame when the PZT stack pre-stress is applied during fabrication, and (v) it must enable the device to maintain high dynamic stability during bone cutting. A simple frame geometry, that importantly includes a cut-through as seen in Fig. 1, enables all these requirements to be met.

Nevertheless, the current design shown in Fig. 1a presents some limitations, as the entire device must be replaced once the blade blunts after repeated bone cutting. One potential solution is to design the cutting blade as a detachable component, fixed to the transducer apex via a threaded stud, allowing only the blade to be replaced. However, experimental results show that a removable blade can introduce slight misalignment with the transducer, leading to an unbalanced dynamic response during cutting. In addition, the mechanical interface increases damping, resulting in heating during excitation, which reduces cutting performance.

The overall size of the surgical device is 10 mm × 16 mm at the base × 45 mm length, which, for example, is compatible with the diameter of the tubular retractors used for microdiscectomy 16–18 mm[37]. The blade has a cutting tip of 7.75 mm in length and 0.5 mm in width. These dimensions create a device that can be tuned to a target resonance frequency comparable with commercial hand-held ultrasonic bone surgery devices (usually in the 20 – 30 kHz range).

**Fig. 1 | Design of the miniature ultrasonic surgical device and characterisation of the electromechanical responses. a** Exploded view of the CAD model showing the components of the device. **b** Front and side views of the fabricated device (the rulers show measurements in millimetres). **c** FEA predicted mode and **d** LDV measured mode, with red to blue indicating high to low displacement. **e** Impedance magnitude and phase characteristics as a function of frequency. **f** Harmonic response from LDV displacement measured at the tip of the blade.

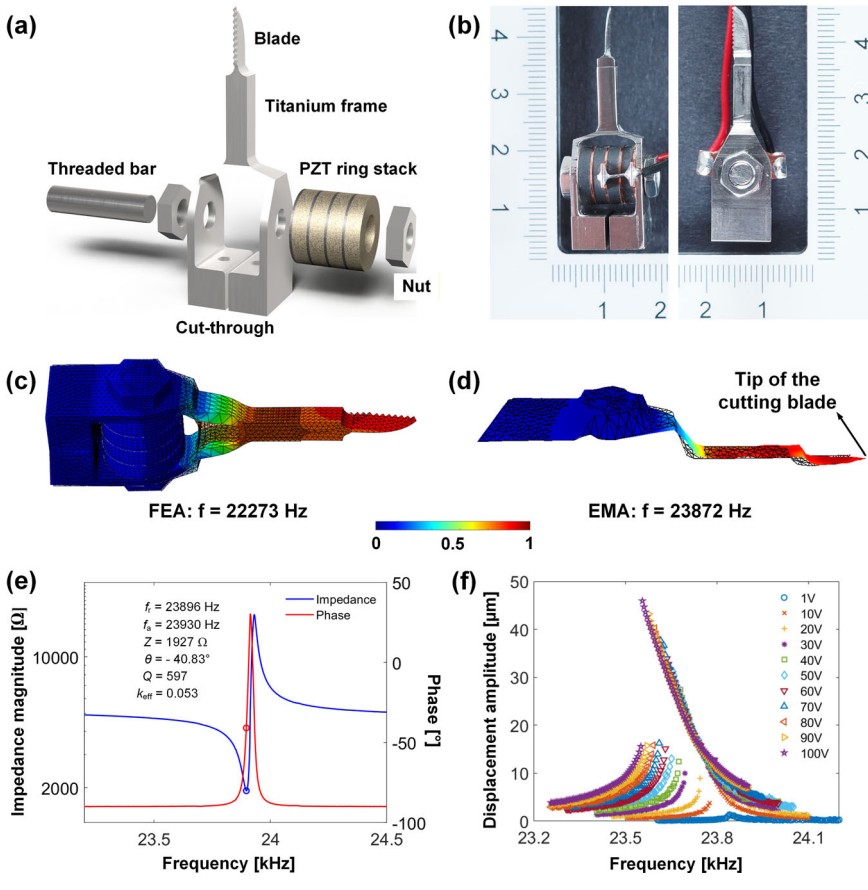

At the operating resonance frequency of the device, the frame amplifies the small axial displacement of the PZT ring stack into a large flexural deformation of the working side of the frame and hence into axial vibrational displacement of the blade. This can be seen in the FEA modelled result of the vibrational mode shape in Fig. 1c which is validated by scanning laser Doppler vibrometer (LDV) measurement of the mode shape in Fig. 1d.

The FEA predicted and experimentally measured mode shapes, which are in close agreement, also show that there is no vibration of the connector side of the transducer and no bending in the PZT stack, which positively affects the electromechanical efficiency of the device[38–40]. The predicted and measured resonance frequencies vary by 7%; $f_r$ is 22,273 Hz from the FEA model and 23,872 Hz from the LDV measurement. This variation is ascribed to the dimensional tolerance of the cut-through in the metal frame, which is found to have a strong effect on frequency when pre-stress is applied to the PZT stack.

From the impedance measurement shown in Fig. 1e, the resonance frequency $f_r$ is measured to be 23,896 Hz and the impedance magnitude and phase at $f_r$ are 1927 Ω and −40.8°, respectively. This highlights a significant challenge for the miniature surgical device; BLTs typically exhibit an impedance magnitude an order of magnitude lower and a phase of 0°. The miniature device, therefore, exhibits capacitive characteristics at resonance and requires an impedance matching circuit and a resonance tracking system. This unusual capacitive behaviour at the series resonance, in contrast to the near-resistive behaviour of the BLT configuration, is attributed to the complex interaction between the mechanical resonance, the electromechanical coupling, and the compliance of the metal shell. These interactions introduce a phase lag between the force and velocity of the device, and significantly reduce the effective electromechanical coupling, contributing to the non-neutralised static capacitance $C_0$ (see Fig. 2) at the series resonance.

The electromechanical coupling coefficient $k_{eff}$ is calculated from the impedance spectrum (Fig. 1e) using Eq. (1)[41], providing a measure of the ultrasonic surgical device's conversion efficiency from electrical energy to mechanical vibration. The mechanical quality factor, $Q$, calculated using the 3 dB method, is an important indicator of the surgical device's potential to achieve large displacement amplitude with low losses.

$$k_{eff}^2 = \frac{f_a^2 - f_r^2}{f_a^2} \qquad (1)$$

The measured coupling coefficient $k_{eff}$ is calculated to be 0.053, which is significantly lower than a conventional BLT that typically has a value of 0.2 to 0.5[42–44]. This low $k_{eff}$ is due to the high impedance magnitude and capacitive characteristics of the device at the resonance frequency. Despite this, the mechanical quality factor $Q$ is almost 600, which is more representative of a BLT that is often in the range of several hundred to a few thousand[45,46].

The result of a harmonic analysis of the device is shown in Fig. 1f. The displacement at the tip of the cutting blade is measured in an upward frequency sweep through the resonance frequency at 5 Hz steps using a 1-D LDV, at increasing increments of excitation voltages. The device exhibits a characteristic nonlinear softening with the backbone curve of the frequency response bending slightly towards the left[47] as excitation voltage is incremented from 1 up to 100 V$_{rms}$. At the highest excitation level, a displacement amplitude of 46 μm peak-to-peak is measured at the tip of the blade, which is known to be compatible with effective bone cutting[4,48].

The characterisation results of the miniature ultrasonic surgical device demonstrate its ability to achieve a large vibration displacement amplitude of the blade tip, 40 μm being sufficient for bone cutting. The good agreement between the FEA model and the experimental measurements for the mode shape validates the design, with minimal unwanted vibrations and no

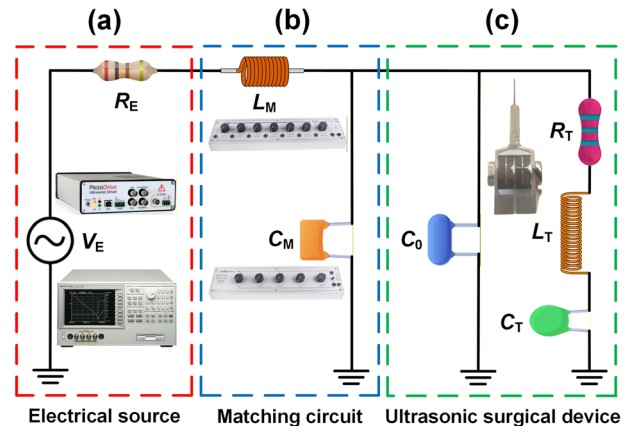

**Fig. 2 | Schematic showing the implementation of an *LC* impedance matching circuit. a** Electrical source. **b** Matching circuit. **c** Ultrasonic surgical device.

bending in the PZT stack. Although the device exhibits a higher impedance magnitude and lower coupling coefficient compared to a BLT, indicating it will require an impedance matching circuit for resonance tracking, the measured mechanical quality factor $Q$ and achievable displacement amplitude at the blade tip indicate that it is a viable alternative configuration for an ultrasonic surgical device.

It should be noted that commercial ultrasonic bone cutting devices integrate a liquid cooling system, often by incorporating an internal channel that allows saline solution to be directed at the blade tip. For future clinically relevant tests of the device, the design will be developed further to incorporate cooling and the flextensional components will be enclosed in a sealed casing to prevent coolant ingress and to protect the electrical components.

## Electrical impedance matching circuit

Due to the high impedance magnitude (1927 Ω) and strongly capacitive (−40.8°) characteristic of the surgical device at the resonance frequency, an impedance matching circuit is required to maximise the energy transfer to the cutting blade. Figure 2 shows a simplified equivalent circuit of the surgical device at resonance. The device is represented by a Butterworth-Van Dyke (BVD) model[49], which is the most commonly used circuit to represent an ultrasonic transducer at the series resonance. Applying the BVD model to the ultrasonic surgical device, $R_T$ represents the radiation and mechanical losses, $L_T$ and $C_T$ are the motion inductance and capacitance, respectively, and $C_0$ is the device's clamped capacitance. The device is excited by an electrical source through an $LC$ impedance matching circuit. $V_E$ represents the electrical source, $R_E$ is the internal resistance of the electrical source, and $L_m$ and $C_m$ are the inductance and capacitance of the impedance matching circuit, respectively.

It is known that losses in an ultrasonic surgical device are excitation-level dependent[45,50], therefore applying optimal impedance matching parameters $L_m$ and $C_m$ derived from measurements at a low-level excitation is unlikely to result in the largest achievable displacement amplitude at the cutting blade when the device is operated at a high excitation level required to cut bone. To explore this, a parametric study is carried out at both low-level excitation, using an impedance analyser, and high-level excitation using a resonance tracking system. The resonance tracker is a control system that continuously adjusts the driving frequency of the transducer to maintain resonance, under the continuously changing loading conditions the device experiences in operation. This includes heating and cutting forces that result in changes to the resonance frequency.

## Impedance matching at low and high excitation levels

Figure 3a, b shows the effect of the two excitation levels on the electromechanical characteristics of the surgical device with an $LC$ impedance matching circuit implemented. Low-level excitation measurements use an impedance analyser at 1 V, and high-level excitation measurements use a

resonance tracking system at 30 V. The ranges of inductance, $L_m$, and capacitance, $C_m$, of the $LC$ impedance matching circuit are chosen to be 0 to 30 mH (with an increment of 2 mH) and 0 to 300 pF (with an increment of 30 pF), respectively. The impedance magnitude and phase, coupling coefficient $k_{eff}$, mechanical quality factor $Q$ and displacement amplitude at the tip of the blade are calculated for each inductance $L_m$ and capacitance $C_m$.

For low-level excitation impedance matching, Fig. 3a, the impedance magnitude drops significantly, from 1.9 kΩ to <100 Ω, as the inductance increases, whereas it is unaffected by the capacitance. A similar trend is observed for the phase, which changes from a capacitive characteristic at a low inductance to a resistive characteristic at the highest inductance. Again, capacitance has little effect. The electromechanical coupling coefficient, $k_{eff}$, increases from 0.05 to almost 0.3, peaking at the highest inductance and capacitance. This is mainly due to a large shift downwards of the resonance frequency as $L_m$ and $C_m$ increase, whereas the anti-resonance frequency is hardly changed. The mechanical $Q$ exhibits its highest value (around 900) when $L_m$ is around 20 mH; in this regime, device losses will be minimised.

For high-level excitation impedance matching, 30 V is selected, which is significantly higher than for the low-level excitation impedance matching but also ensures the surgical device is excited in a linear regime. From Fig. 3(b), the impedance magnitude at steady-state vibration of the device is around 4 kΩ when $L_m = 0$ and maintains this value until $C_m > 240$ pF when it increases up to 6.5 kΩ. In general, the impedance magnitude decreases at a high rate as $L_m$ increases, whereas there is little effect of $C_m$. From measurements of the blade tip displacement, a peak amplitude of around 30 μm peak-to-peak is excited when the matching inductance is close to 20 mH, and it rapidly decreases on either side of 20 mH. The maximum displacement is identified when $L_m = 20$ mH and $C_m = 270$ pF. This set of matching parameters is therefore adopted for the matching circuit to investigate the relationship between impedance magnitude, displacement amplitude and excitation voltage.

The results are shown in Fig. 3c. The impedance magnitude grows quadratically from 250 to 420 Ω, as the applied voltage is increased from 10 to 90 V in increments of 10 V. The optimal output impedance magnitude of the resonance tracking system is 400 Ω, meaning that the maximum energy transfer will be achieved if the impedance magnitude of the miniature surgical device driven continuously for bone cutting is close to this value. A linear increase in the displacement amplitude of the blade is observed, reaching 60 μm peak-to-peak at 90 V. However, it should be noted that this is not a limit of the ultrasonic surgical device, but rather a limitation of the LDV measurement set-up.

Even though the optimal set of impedance matching parameters has been identified ($L_m = 20$ mH and $C_m = 270$ pF) from the high-level excitation matching, the impedance magnitude of the device has also increased significantly (from 250 to 420 Ω), highlighting the influence of excitation level on impedance. Additionally, device loading during bone cutting will likely result in a further increase in impedance, leading to a reduction in efficiency. Therefore, to understand how the resonance tracking and impedance matching circuit perform during bone cutting, experiments were focused on the effects of cutting speed, penetration rate, stroke, and total cutting depth.

For all in vitro bone cutting experiments, the optimal impedance matching parameters ($L_m = 20$ mH and $C_m = 270$ pF) remain unchanged, despite the variations in the mechanical loading and excitation conditions. To maintain resonance, the control system was operated in constant-voltage mode and tracked the series resonance by locking the voltage and current phase of the device to a 0° reference angle. The phase-control feedback gain was set to 0.01, which recovered the stability of the system from detuning within approximately 100 ms during bone cutting.

## Ultrasonic bone cutting experiments

Prior to integration with the Kuka robot, a dedicated test rig was developed to perform controlled in vitro cutting tests on bone samples, as shown in detail in Fig. 4. The test rig is configured to measure the following parameters: lateral cutting force and bone penetration force, impedance magnitude of the surgical device and consumed power of the resonance tracking

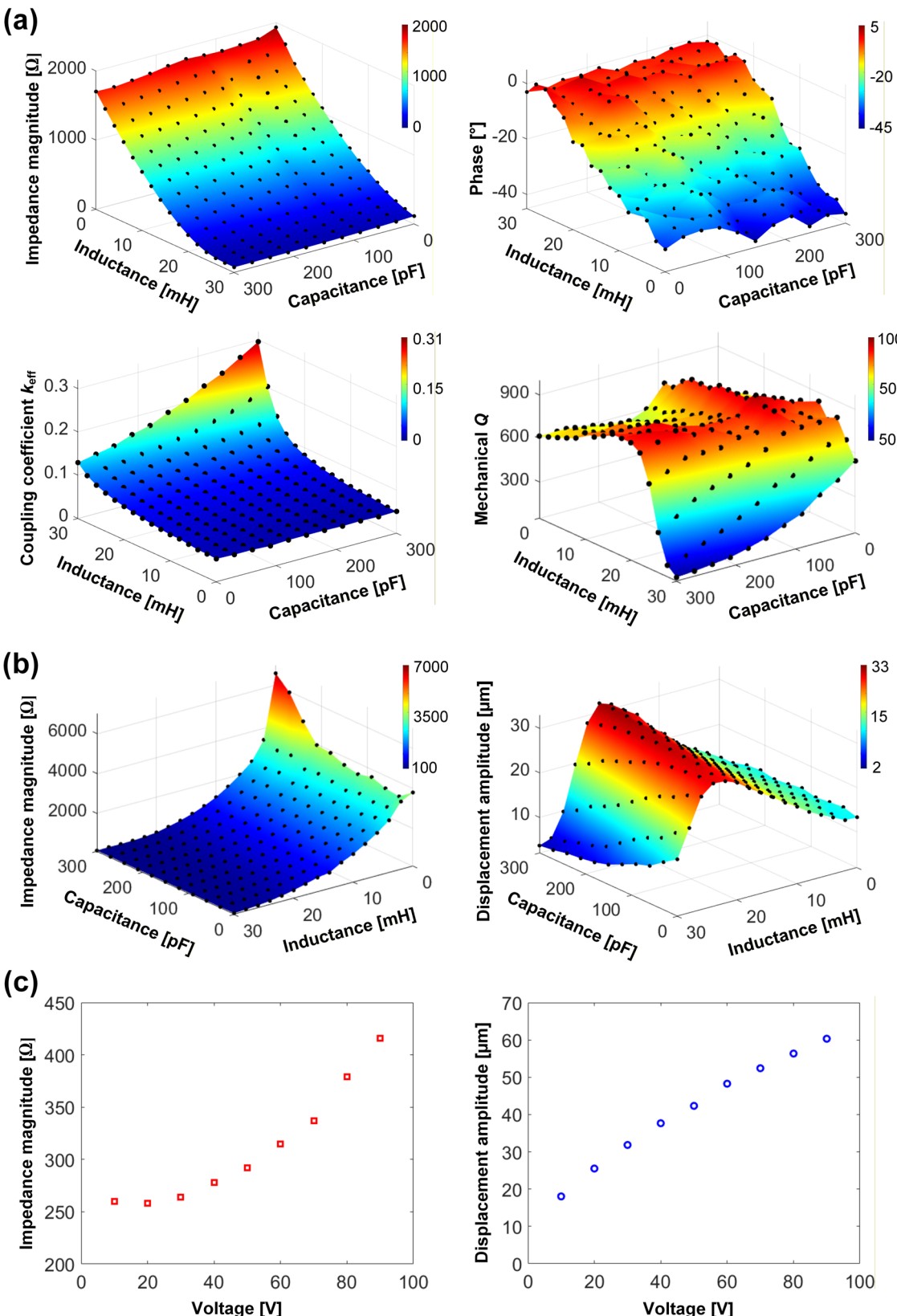

**Fig. 3 | Parametric study of the matching inductance and capacitance on the electromechanical responses of the miniature ultrasonic surgical device.**
**a** Impedance magnitude, phase, electromechanical coupling coefficient k$_{eff}$, and mechanical Q dependence on the matching inductance $L_m$ and capacitance $C_m$, measured by an impedance analyser at 1 V excitation. **b** Impedance magnitude and displacement amplitude measured at the tip of the blade dependence on the matching inductance $L_m$ and capacitance $C_m$, excited at 30 V with a resonance tracking system. **c** Impedance magnitude and displacement amplitude at the tip of the blade of the surgical device for increasing voltage from 10 to 90 V, adopting optimal impedance matching parameters $L_m$ = 20 mH and $C_m$ = 270 pF.

**Fig. 4 | Test rig for ultrasonic tissue cutting experiments. a** Schematic showing test rig components. **b** Image of the test rig. **c** Close-up view of the linear actuator and prototype bone surgery device showing the directions of the cutting forces, $F_x$ and $F_z$, cutting speed, $V_L$, stroke length, $L$, penetration rate, $V_z$, and target penetration depth, $D$.

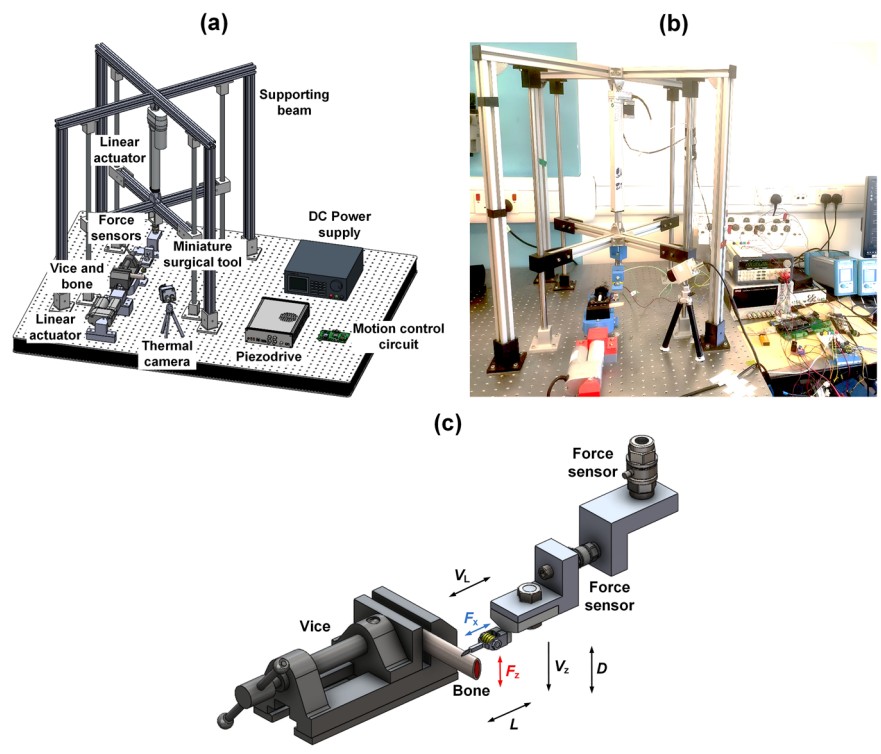

system, and temperature at both the cut site and the piezoelectric stack. Fresh porcine rib was used for the tissue cutting experiments because of its similarity in density, porosity, microstructure, and composition to human bones[51–53]. For all tests, the cutting speed, $V_L$, is set at 3 mm/s, which guarantees cutting stability and control within the accuracy of the test rig. The stroke, $L$, is 2 mm and the target depth, $D$, is 2 mm, selected as commensurate with the size of the dimensions of the bone sample. The vertical engagement speed, which is the device penetration rate, $V_z$, is set to 5, 10 and 15 µm/s, as these maintain sufficiently low cutting force in the penetration direction to be able to maintain control of the test rig stability.

The sawing motion is chosen as a simple, repeatable and controllable experimental protocol that ensures stable blade and bone interaction, enabling systematic comparison of cutting force, temperature, and electrical characteristics. This motion is not intended to mimic the way a clinician performs ultrasonic osteotomy and is not intended to be a protocol for robotic surgery, but should be regarded as a laboratory-based evaluation method to assess the device's cutting performance. The blade tip displacements are set to 0, 30 and 60 µm peak-to-peak. It should be noted that a newly fabricated miniature ultrasonic surgical device was used for these bone cutting experiments. Each experiment for a given set of parameters was performed only once, primarily to avoid the influence of tool wear, as degradation of the serrated teeth could otherwise affect the measurements.

The results are shown in Fig. 5. It is observed that for the larger amplitude of 60 µm a significant reduction in both the lateral cutting force $F_x$ and the vertical penetration force $F_z$ are achieved compared to the non-ultrasonic cutting. This reduction is particularly noticeable at the slowest penetration rate, $V_z = 5$ µm/s, where the maximum force remains below 1 N, highlighting the advantage of using a higher blade displacement amplitude.

At a higher penetration rate of 10 µm/s, the significant force reduction is lost, however 60 µm amplitude still achieves a 50% reduction in both force components compared to non-ultrasonic cutting, staying under 4 N. This suggests that at this penetration rate, the bone removal efficiency decreases, especially when the blade engages deeper within the bone. Furthermore, 30 µm amplitude results exhibit similar or even larger forces than the non-ultrasonic forces, indicating that a lower amplitude at this penetration rate provides no advantage.

When the penetration rate is increased to 15 µm/s, the advantage of ultrasonic cutting is completely lost. At 60 µm amplitude, the sudden rise in $F_x$ at 65 seconds suggests a temporary loss of resonance tracking, followed by oscillitory characteristics, indicating the system attempts to restore the amplitude as the blade engages deeper within the bone. This leads $F_z$ to rise sharply due to increased cutting depth. Unexpectedly, 30 µm amplitude results in lower forces than both the 60 µm amplitude and the non-ultrasonic cutting, with values even lower than those observed at penetration rates of 5 and 10 µm/s. This may be attributed to cutting occurring in a region of the cortical bone where mechanical properties are more favourable, which identifies an issue in the test, of inconsistency of bone properties. Also, the porcine rib samples used in the cutting experiments exhibit natural variation in cross-sectional geometry, which will lead to differences between tests in the number of blade teeth engaged with the bone at any given penetration depth.

The impedance magnitude for 30 µm amplitude starts at approximately 100 Ω, lower than for 60 µm. Additionally, the starting power difference is about 2 W, highlighting the influence of the excitation level. For all engagement rates, both displacement amplitudes result in a sudden increase in impedance magnitude and a decrease in power (more pronounced for 60 µm than 30 µm). These changes occur at around 200, 75, and 40 seconds for 5, 10, and 15 µm/s engagement rates, respectively. This is due to the full engagement of all the serrated teeth of the blade with the bone, increasing the surface contact area. Under these conditions, the resonance tracking system was unable to maintain stability.

Unlike the 5 µm/s penetration rate, the other two penetration rates exhibit noticeable oscillatory behaviour at a frequency much lower than the cutting speed, as the blade penetrates deeper into bone. These oscillations start at 110 seconds for the 10 µm/s penetration rate and at 65 seconds for the 15 µm/s penetration rate. This clearly indicates a temporary loss of resonance, with the tracking system attempting to recover it. The thermal response further supports these findings; for 60 µm and 5 µm/s the temperature steadily increases as the blade reaches full depth, indicating a strong vibro-impact micro-sawing interaction between the blade and bone[54–56]. In contrast, for both 10 and 15 µm/s penetration rates, temperature decreases occur at times corresponding to the changes in impedance and power,

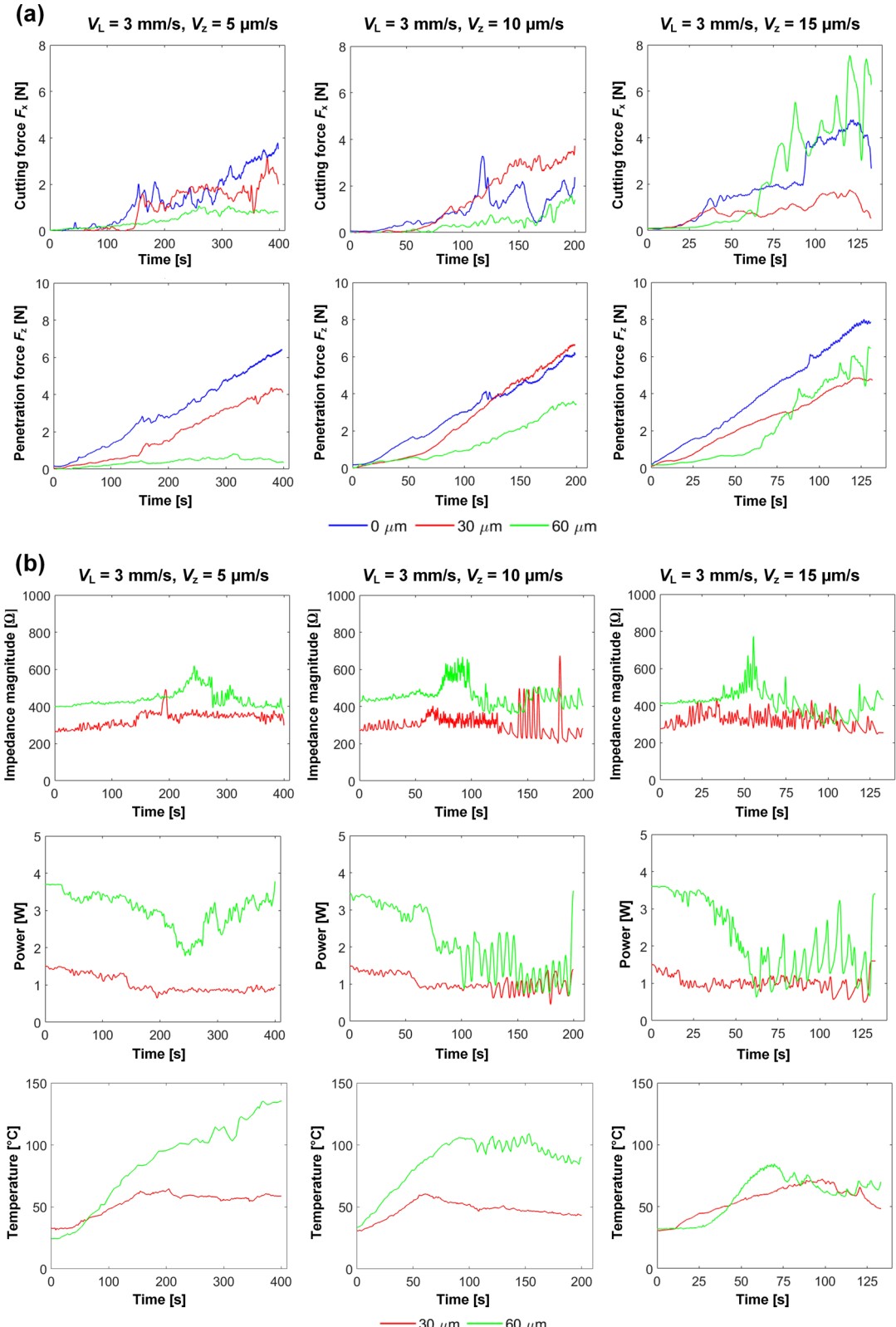

**Fig. 5 | Ultrasonic tissue cutting results on the test rig. a** Forces $F_x$ and $F_z$ at displacement amplitudes 0, 30 and 60 µm and penetration rates 5, 10 and 15 µm/s. **b** Impedance magnitude, power consumption, and temperature measured at the cut site at amplitudes of 30 and 60 µm and penetration rates 5, 10 and 15 µm/s.

reflecting a reduced vibro-impact micro-sawing interaction between the blade and bone.

Figure 6 shows microscopic images of the bone cuts made at 5 µm/s penetration rate with increasing blade displacement amplitude, while the table summarises cutting depths for all tested parameters. For this

test-rig set-up, it is optimal to operate the device at a high displacement amplitude and slow penetration rate. This approach generates extremely low forces and stable electrical responses, evidencing a strong vibro-impact micro-sawing interaction between the blade and bone. The high temperature also indicates strong blade and bone interaction and is

**Fig. 6 | Microscopic images of the dissected porcine bone samples and depths of cut. a** At three displacement amplitudes 0, 30 and 60 µm. **b** At three penetration rates 5, 10 and 15 µm/s.

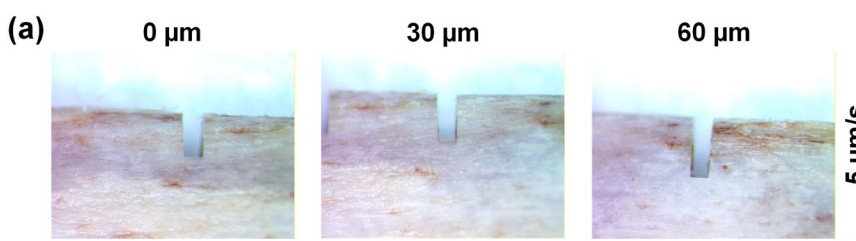

**(a)**

| 0 µm | 30 µm | 60 µm |

5 µm/s

1mm

**(b)**

| Depth of penetration [mm] when varying blade amplitude and penetration rate $V_z$ | | | |
|---|---|---|---|
| | | Blade amplitude [µm] | |
| Penetration rate [µm/s] | 0 | 30 | 60 |
| 5 | 0.84 | 1.48 | 2.04 |
| 10 | 0.71 | 1.04 | 1.35 |
| 15 | 0.59 | 0.87 | 1.04 |

**Fig. 7 | Integration of the miniature ultrasonic surgical device with the Kuka robot for bone cutting tests. a** Schematic of the robotic cutting platform. **b** Image of the experimental set-up. **c** Close-up view of the device integrated with the robot showing directions of the cutting forces, $F_x$, $F_y$ and $F_z$, cutting speed, $V_L$, stroke length, $L$, cut depth increment $\Delta_z$, and target penetration depth $D$.

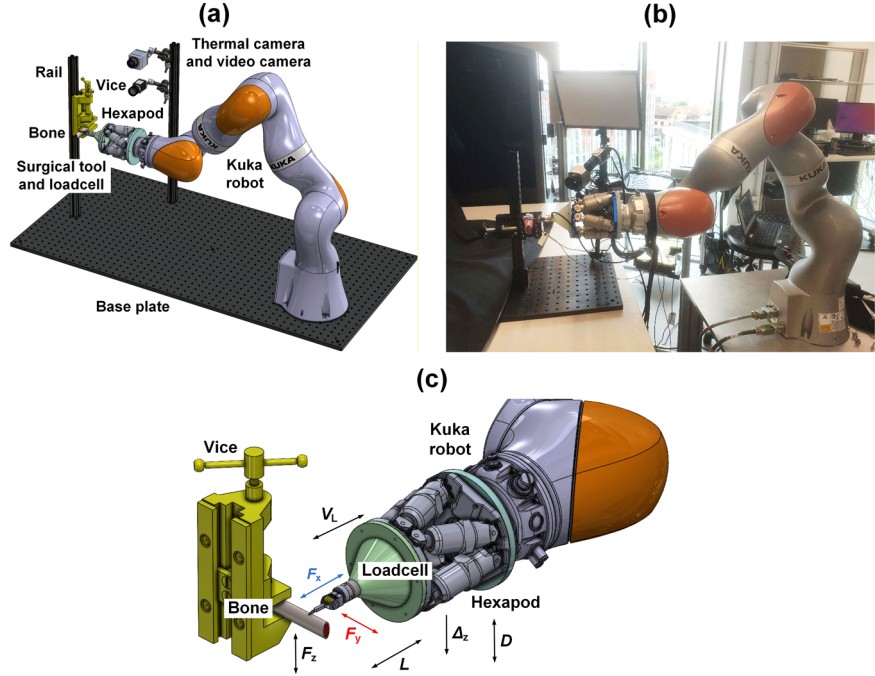

generally mitigated in ultrasonic bone surgery devices by incorporating cooling.

## Device integration with a Kuka robot

From the findings of tests in the test rig, 60 µm blade amplitude and 5 µm/s penetration rate are selected for testing the miniature device on the Kuka robot. The set-up for in vitro robotically assisted bone cutting experiments is shown in Fig. 7. Cutting experiments are performed with cutting speed, $V_L$, set at 2, 3, and 4 mm/s, depth increment, $\Delta_z$, at 20 and 40 µm per stroke, and target depth, $D$, at 1 and 1.5 mm. Again, for these experiments, only one cutting test was performed for each set of given parameters. The results are presented in Fig. 8.

All three measured force components, $F_x$, $F_y$, and $F_z$, remain below 1 N, despite increases in cutting speed, depth increment per stroke, and target depth. The lateral force $F_y$ is insignificant (close to 0 N) across all cutting parameters, suggesting that the increased freedom and compliance of the motion-driving mechanism (Kuka robot) allow the miniature device to create a cut slightly wider than the blade itself, minimising friction with the bone. This could also mean that no significant bending motion occurs in the blade.

At higher cutting speeds (3 and 4 mm/s), the vertical penetration force, $F_z$, is less than 0.2 N. This suggests that the combined vibro-impact and micro-sawing motion of the blade, superimposed onto the reciprocal sawing speed, enhances the dynamic interaction with the bone, promoting fracture and increasing bone material removal rate. Meanwhile, the horizontal cutting force, $F_x$, steadily rises as the blade engages deeper with the bone, reaching approximately 0.5 N. This indicates that the teeth of the blade are effectively removing bone material with each incremental depth increase per stroke.

The impedance magnitude starts at 370 Ω and increases significantly when the device operates at higher speeds and larger incremental depths per stroke. This increase leads to a drop in the acoustic power from an initial 4 to 3 W. A more pronounced change in impedance magnitude is observed at 250 seconds, for a cutting speed of 3 mm/s, an incremental depth 20 µm, and a target depth 1.5 mm. Under these conditions, power fluctuations occur, temporarily dropping to nearly 2 W. This is likely to occur when the blade encounters a harder bone region, due to the material properties of the bone not being constant over the cut site.

The temperature at the PZT stack gradually increases to approximately 60 °C for all cutting parameters. At the cutting site, the highest temperature is recorded for the highest cutting speed, which generates higher friction.

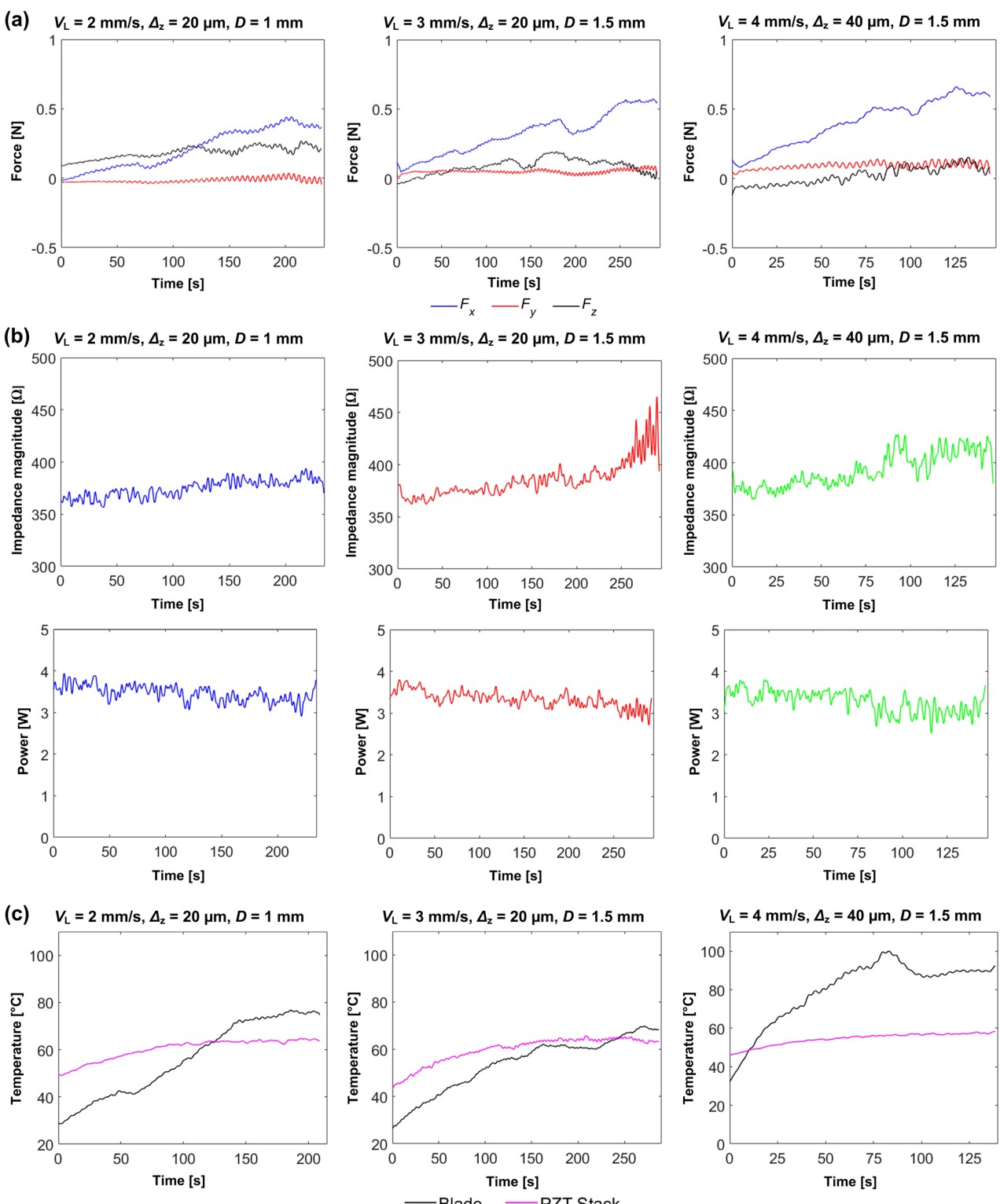

**Fig. 8 | Ultrasonic tissue cutting results facilitated with the Kuka robot: a** Force $F_x$, $F_y$ and $F_z$. **b** Change in impedance magnitude and power consumption. **c** Temperature measured at the piezoelectric stack and the cutting site at three cutting speeds 2, 3 and 4 mm/s and two depth increments per cycle 20 and 40 μm.

Nonetheless, a sudden drop in the measured cutting site temperature is observed at approximately 80 seconds, which is not reflected by corresponding changes in the force, impedance magnitude, or power. This behaviour is likely attributed to the presence of ejected bone debris at the high cutting speed, which alters the surface emissivity and affects the temperature measurement, given that the emissivity was calibrated for the Ti-6Al4V blade.

Microscope images and bone cut profiles are shown in Fig. 9. The cut edges remain intact, with no visible burrs or microdamage for all three parameter sets. Additionally, the blade has nearly fully penetrated the cortical layer at the 1.5 mm target depth, as indicated by the darker colouration at the bottom of the cut, which signifies marrow exposure. The measured cut depths are 0.78, 1.21, and 1.42 mm, while the corresponding widths are 0.56, 0.53, and 0.54 mm, for all three cutting parameter sets.

**Fig. 9 | Cuts in the porcine bone sample at three cutting speeds, 2, 3 and 4 mm/s and two depth increments per cycle, 20 and 40 μm. a** Microscopic image. **b** Measurement.

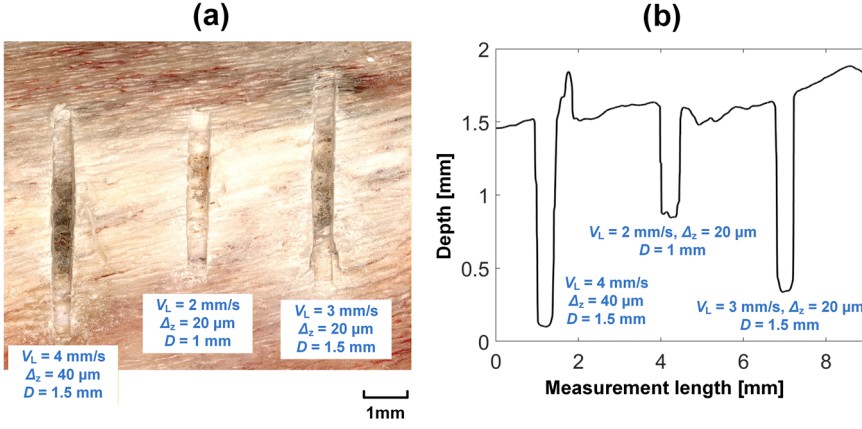

Results from the in vitro bone cutting experiments with the Kuka robot differ from the observations made using the test rig. This highlights the advantages of the high precision and additional degrees-of-freedom, which enable compliant interactions between the blade and bone. This setup provides more controlled engagement and, therefore, accuracy, which is difficult to achieve using the rigid test rig.

A higher cutting speed and greater incremental depth per stroke enhance the interaction between the blade and bone, leading to increased temperature at the cutting site. This highlights the need for cooling, which is typically incorporated in ultrasonic surgical devices. Future research will focus on the optimisation of blade geometries, tailored to specific surgical procedures.

Additionally, integrating haptic mechanisms with force feedback strategies could improve the motion control of the robot, enabling more precise bone cutting along pre-defined pathways.

In summary, the in vitro bone cutting performance of the miniature ultrasonic surgical device, demonstrated using both the test rig and the Kuka robot, can be attributed to three main factors. First, the proposed transducer architecture enables effective miniaturisation while maintaining high electromechanical coupling and displacement amplification at the blade at a clinically relevant resonance frequency, overcoming size limitations associated with BLT-based devices[48,57] and conventional classes of flextensional transducers[31,33,34]. Second, electrical impedance matching using an *LC* network combined with tracking of the series resonance with a phase-lock loop under varying mechanical loads, maximises energy transfer and stabilises displacement of the blade against load- and excitation-induced frequency shifts[58]. Further improvements may be achieved through adaptive impedance matching capable of real-time tuning of the matching elements[59]. Finally, the simple reciprocal sawing motion with slow penetration, together with the serrated blade, has resulted in very low cutting forces; however, optimisation of blade geometry for specific surgical applications is necessary to perform clinically relevant device performance tests[60].

## Methods
### Fabrication of the surgical device
Four hard piezoelectric PZT rings (PIC-181, PI Ceramic, Germany) of dimension outer diameter 10 mm, inner diameter 5 mm and thickness 2 mm are used, with material properties defined in ref. 4. Four copper electrodes, one metal frame made from titanium grade 5 alloy Ti-6Al4V, one threaded bar and two nuts made from A4 tool steel are all thoroughly cleaned using isopropyl alcohol (IPA) solution prior to the fabrication process. The PZT ring stack is then assembled and inserted into the metal frame, before the threaded bar and two nuts are installed (see Fig. 1a). Two wires are attached to the live and ground terminals of the PZT stack. Torque is gradually applied to the nuts and threaded bar, increasing from 0.5 to 3.0 Nm in increments of 0.5 Nm, to achieve an ultimate pre-stress of around 30 MPa[61]. Electrical impedance is recorded for each applied torque

increment during fabrication, with a stabilisation of resonance frequency and impedance magnitude being an indicator of sufficient pre-stress. The device is then allowed to settle for a few weeks, allowing for the electrical characteristics to reach a steady-state, confirmed by no further change in the impedance.

### Electromechanical characteristics analysis
The impedance characteristics of the miniature ultrasonic surgical device are measured using an impedance analyser (4294A, Agilent, USA) with 1 V swept signal applied across the bandwidth of the resonance frequency of the device.

Experimental modal analysis of the surgical device is conducted using an MSA-100 3-D laser Doppler vibrometer (Polytec, Germany), and the results are compared with the vibration mode shapes predicted in finite element analysis (FEA) using Abaqus-Simulia (Dassault Systèmes, France) software.

Harmonic analysis experiments are performed to understand how the ultrasonic displacement amplitude at the tip of the blade varies with excitation level. The surgical device is excited with a frequency sweep from below to above the resonance frequency. A burst sine wave is used, generated by a signal generator (Agilent 33210A, USA) and amplified by a power amplifier (HFVA-62, China). The vibration displacement response at the tip of the blade is measured using a 1-D laser Doppler vibrometer (OFV 303, Polytec, Germany).

### Design of the electrical impedance matching circuit
The impedance analyser (4294A, Agilent, USA) which has an output impedance of 50 Ω was used for the low-level excitation analysis. For high-level excitation, a resonance tracking system (PDUS210-800, Piezodrive, Australia) is employed with an optimal output impedance magnitude of 400 Ω. For both excitation regimes, an inductance decade box (DL07, ELC, France) and a capacitance decade box (DC05, ELC, France) are used for the parametric study.

### Bone cutting test rig
Figure 4 shows the ultrasonic tissue cutting test rig. The surgical device is fixed to a force sensor (9311b, Kistler, Switzerland), which is attached horizontally to an L-shape plate, which is then connected to another force sensor (9321b, Kistler, Switzerland) via an L-shape plate that is deployed in the vertical direction (see Fig. 4c). The assembly is then fixed to the cross-beams which are driven by a stepper motor controlled linear actuator (GLA750-STEP-20-3-285-390, stroke length 285 mm, Gimson Robotics, UK), which is mounted between the table and the upper fixed crossbeams. Device penetration rate in the vertical direction, denoted as $V_z$, is facilitated with a bespoke motion control circuit, and the target depth, *D*, can also be pre-defined. The crossbeams move in a purely vertical direction with the support of four linear rails and four embedded needle bearings to minimise lateral motions of the surgical device.

A fresh porcine rib bone is held firmly in a vice, as shown in Fig. 4c, which is mounted to another DC driven linear actuator (GLA750-P, stroke length 100 mm, Gimson Robotics, UK) that drives the bone in a sawing action at a speed $V_L$ and with a stroke length $L$ in the horizontal direction. The cutting forces in both horizontal direction, $F_x$, and vertical direction, $F_z$, are measured by the two force sensors simultaneously. Before each cutting experiment, the static offsets of both force sensors were reset to zero in the charge amplifiers (5015a, Kistler, Switzerland), effectively removing the influence of the mass of the device and weight of the fixture. Minor drift in the force measurement caused by ambient temperature variations and charge leakage was corrected using linear line fitting compensation. Additionally, a thermal camera (TIM 160, Micro-Epsilon, Germany) is mounted close to the cutting site, focusing on the tip of the blade to record the cutting temperature. The bone cuts are measured by a microscope (AmScope, USA).

### Robotic bone cutting platform

The experimental arrangement of the integration of the miniature ultrasonic surgical device with a Kuka robot is presented in Fig. 7.

The ultrasonic surgical device is attached to a six-axis loadcell (Nano17 SI-25-0.25, ATI Industrial Automation, USA) via an aluminium fixture to measure the forces in the cutting direction ($F_x$), the transverse direction ($F_y$), and the engagement direction ($F_z$). The device and loadcell are fixed to a six degree-of-freedom positioning hexapod (Solano, Symétrie, France), which is then connected to the flange of a Kuka robot (LBR iiwa 14 R820, Kuka AG, Germany), capable of seven axes of motion. The hexapod is employed due to its high spatial accuracy (±0.1 μm), which is significantly better than the experimentally confirmed repeatability of the Kuka robot arm alone (±0.05–0.1 mm), and is therefore more suitable for testing the miniature ultrasonic surgical device. For comparison, the operating precision of medical robots is in the range ±0.1–0.15 mm; the Kuka robot's is approximately ±0.15 mm[62].

The entire assembly is mounted to a bench, and a vice is fixed to a metal extrude that is bolted vertically to the bench, which is used to clamp a fresh porcine rib bone. A thermal camera (TIM 640, Micro-Epsilon, Germany) and a high-resolution video camera (a2A1920-160 μcBAS, ace 2R, Basler, Germany) are mounted to another metal extrude to record the cutting temperature and video each test. The dissected bone sample is analysed using a digital microscope (VHX-6000, Keyence, Japan).

During operation, the surgical device, loadcell and hexapod assembly is deployed by the robot from a 'home' position (at some distance from the cutting site) to the 'ready-to-cut' position (with the middle tooth of the blade serrations aligned and in contact with the bone). Thereafter, the surgical device is powered, and the high precision hexapod is activated to perform bone cutting. The sequence of cutting is that once the surgical device is deployed at the 'ready-to-cut' position, the robot will drop 500 μm to ensure the device is in contact with the bone, which is associated with a small increase in the force $F_z$. The linear motion of the hexapod is activated, and the surgical device is driven forward at speed $V_L$ for half of the stroke length $L$ and backward for the full stroke length, followed by forward for other half, returning to the initial position. Next, the hexapod lowers the surgical device by one depth increment $\Delta_z$, and the cutting cycle repeats until the target depth, $D$, is reached.

### Bone sample preparation

Fresh porcine ribs are procured and excarnated using a scalpel and are stored in isotonic phosphate buffered saline (PBS) to maintain hydration. The bone sample is used in tests on the same day as procurement.

### Data availability

The data that support the findings of this study are available from the corresponding author upon reasonable request.

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

## Acknowledgements

This work was supported by an EPSRC Programme Grant, Ultrasurge – Surgery enabled by ultrasonics, EP/R045291/1.

## Author contributions

X. Li led the technical development of the study, including designing the methodology, developing and validating the devices, performing formal analysis, conducting experiments, curating the data, drafting the original manuscript, and editing the revision. D. Jones supported methodological design, software implementation for Kuka robot, validation, experimental investigation, data processing, and visualization. P. Valdastri and M. Lucas provided resources and guidance, supervised the direction of the project, contributed to critical revision of the manuscript, and secured funding. M. Lucas additionally contributed to drafting the original manuscript.

## Competing interests

The authors declare no competing interests.
