## [Transparent Peer Review file · Communications Engineering]

A miniature ultrasonic surgical device based on a flextensional configuration with a pre-stressed piezoelectric stack

Corresponding Author: Dr Xuan Li

Version 0:

Reviewer comments:

Reviewer #1

(Remarks to the Author)

Overall Evaluation:

This paper presents an innovative design for a miniature ultrasonic surgery device based on a flextensional configuration, in which a pre-stressed stack of piezoelectric rings serves as the ultrasonic vibration source. The prototypes demonstrate the ability to sustain high excitation levels for extended periods without showing signs of performance degradation. Although the radial dimension of the proposed design remains somewhat larger than the typical outer diameter (10–12 mm) of existing robotic instruments, this work offers a promising alternative configuration for the miniaturization of surgical ultrasonic devices.

Major Concerns:

- 1) This study motivates the design by stating “Ultrasonic bone scalpels are known to offer benefits” as well as “all current commercial devices are too large to be integrated with the flexible endo-wrist of a surgical robot”. However, the authors did not really propose a design with the wrist. And it is not clear in what clinical scenario, a bone scalpel with an endo-wrist is needed.
- 2) The proposed miniature ultrasonic surgical device has a base size of 10 mm × 16 mm. It should be clarified whether this overall size is compatible with the anatomical constraints typically encountered in bone surgery or in endoscopic surgery.
- 3) The authors presented the design without any design optimization. Structure parameters may be optimized.
- 4) A comparative analysis between the proposed flextensional ultrasonic device and a longitudinal bolted Langevin transducer (BLT) of similar base dimensions is certainly welcome and will be highly appreciated to differentiate the true values of the two designs.
- 5) The issue of sealing design can be addressed or at least discussed to ensure the electrical components are protected from liquids in clinical environments. Typically, an outer casing is used to provide sealing, and it should contact the device at a zero-displacement plane (vibration node) to minimize attenuation of output displacement. The current configuration appears to have only one such node at the base, which is intended for connection to a surgical robot. How adequate sealing can be achieved without significantly compromising displacement output requires further consideration and justification.

Reviewer #2

(Remarks to the Author)

Congratulations to authors on a paper on an innovative miniature ultrasonic surgical device based on flex tensional configuration .

The concept would benefit integration of Ultrasonic scalpel to robotics & will be very useful in spine surgery .

Reviewer #3

(Remarks to the Author)

It is recognized that authors well-conceived, executed, and documented effort to realise a miniature flextensional ultrasonic surgical device using a pre-stressed PZT ring stack. The manuscript presents a clear design rationale, validates mode shapes via FEA and LDV, investigates impedance matching both at low and high drive levels, and demonstrates promising in-vitro bone cutting results with a Kuka robot. The topic is timely and relevant to robotic minimally invasive surgery. I recommend minor revisions to address several clarifications, figure labelling, and experimental reporting issues listed below.

1. The introduction lacks a clear chapter heading. Please add a heading (e.g., "Introduction") and ensure the introductory paragraphs explicitly state the paper's objectives and the novelty relative to BLT and other flextensional approaches.
2. The "cut-through" in the frame is referred to in the text as critical for the mode-shape and for preventing bending under pre-stress, but it is not explicitly labelled in Fig. 1. Please mark the cut-through on the image.
3. The manuscript reports lateral (F_x) and penetration (F_z) forces measured with force sensors. Please explicitly state how the static offset due to the mass of the device/fixture mounted on the sensor was accounted for (tare/calibration). Provide details: was a zeroing performed before each test? Was any drift observed during runs (and how was it compensated)?
4. A sentence in Section 4 states: "At the higher penetration rate, the significant force reduction is lost." This is ambiguous. Please replace with an explicit quantitative statement: e.g., "At $V_z = 10 \mu\text{m/s}$, the significant force reduction is lost."
5. In Fig. 8(a), some F_z traces fall below zero. Please clarify the sign convention used for F_z (is negative tension or upward pull?) and explain the physical reason for negative readings.
6. For the case $V = 4 \text{ mm/s}$, $\Delta z = 40 \mu\text{m}$, $D = 1.5 \text{ mm}$ the blade temperature shows a sharp rise followed by a decrease. Please explain this behaviour: change in contact area as more serrations engage, or thermal camera field-of-view/ emissivity artefacts. Provide corroborating traces (impedance, power, force) aligned with the temperature trace to support your explanation. If emissivity changes (e.g., bone debris) could affect the reading, mention this limitation.
7. The parametric matching study is thorough, but the paper should more clearly state whether the chosen matching values ($L_m=20 \text{ mH}$, $C_m=270 \text{ pF}$) were final for all in-vitro experiments and whether matching was re-tuned when the device encountered loading variations. Please describe the resonance-tracking controller gains/settings and how fast it can recover from detuning during cutting.
8. I found several small typographical issues (e.g., "noticable" → "noticeable"; "Uder" → "Under" in the test-rig results section). Please perform a careful proofread before final submission.

Reviewer #4

(Remarks to the Author)

Reviews on Communications Engineering

COMMS-25-0581-T: A miniature ultrasonic surgical device based on a flextensional configuration with a pre-stressed PZT stack

This article presents a novel miniaturized design of a flexible ultrasonic transducer for bone cutting. The transducer exhibits promising electromechanical behavior and achieves displacement amplitudes of up to $60 \mu\text{m}$ (pk-pk). In addition, in vitro cutting tests were performed to demonstrate its capability to cut porcine bone. The miniaturization of ultrasonic transducers is an important topic in order to integrate this technology into robot-assisted surgery and thereby perform high-precision and tissue-preserving procedures.

Overall, the manuscript is well written, and the results are novel and significant for scientists in this specific field. However, the data and, in particular, the sample size are limited, which affects the validity of the conclusion. I therefore recommend revising the manuscript to address the following key concerns. Further minor comments and recommendations can be found in the attached document.

1. The article is about a miniature ultrasonic surgical device based on a flextensional configuration with a pre-stressed PZT stack (as per the title). In addition, the article specifically describes two test modalities with two different test setups. It is a valid approach to test the feasibility and functionality of a surgical instrument in the target application (bone cutting). However, it would be beneficial to also have a commercially available device or a traditional Langevin transducer as a comparison in the same test environment to clearly demonstrate the added value of the novel concept. Although it is generally demonstrated that the present concept has potential, it is not entirely clear which advantages can be attributed to the novel transducer concept, the impedance matching and the corresponding resonance tracking, or the shape of the cutting blade. I therefore recommend supplementing the discussion of these points by taking relevant literature into account.
2. The section "Results and Discussion" contains a lot of information on the methodology and the materials used (e.g. test rigs). These are described in more detail in the section "Methods". To avoid redundancy, I recommend describing all test setups and characterization methods in detail only in the chapter on methods.
3. Is the proposed flextensional design suitable for use in minimally invasive robot-assisted surgery? The introduction mentions that traditional transducers (BLT) are not suitable for this purpose due to their size. However, there is also the

option of using an extension or a long cutting blade. What would be the limitations in this case? Furthermore, does the proposed flexural transducer fit through a trocar? This could be explored in more detail in the discussion.

4. The description of the cutting blade design requires further explanation: What are the design considerations for the cutting blade? It appears to be quite different from commercially available ultrasonic osteotomy blades. In addition, why was the blade not designed as a separate part with, for example, a thread? For the target application, the entire transducer would be a disposable product. This can also be mentioned as a limitation in the discussion.

5. "Surgical robot": Is the Kuka robot used a surgical robot? According to the Kuka website, the LBR iiwa 14 R820 model is a collaborative industrial robot. However, Kuka also offers surgical robots, but in this case, these would be the LBR Med 7 R800 or LBR Med 14 R820 models. It would be advisable to rename the "surgical robot" or clearly describe that a test setup with an industrial robot is being used and how it differs from the version used for surgical procedures.

6. The description of the test rig and the cutting process is clear and comprehensible. However, it is not entirely clear why this type of sawing motion was chosen for the test, as this is rarely the case when using ultrasonic osteotomy devices. Could a more detailed explanation be provided in this regard? Furthermore, such a sawing motion and penetration rate are difficult to implement in robot-assisted surgery and are likely to be too slow and imprecise for clinical requirements.

7. Results: Is there an explanation for the fact that at 10 $\mu\text{m/s}$, the forces at 30 μm are even higher than with the non-activated transducer, and then at 15 $\mu\text{m/s}$ they are lower compared to 0 μm and 60 μm or lower cutting speeds? One possible reason, as also mentioned later in the corresponding paragraph, could be that the material tested is biological samples, which by nature exhibit a high variation in mechanical properties (e.g. thickness of cortical bone). There are also synthetic bone models that may allow for more standardized testing of device performance. Furthermore, were the measurements repeated? If so, what is the sample size and what is the standard deviation (statistical analysis)? If not, further measurements on biological or synthetic samples may be necessary in order to make a proper assessment.

Version 1:

Reviewer comments:

Reviewer #3

(Remarks to the Author)

Reviewer #4

(Remarks to the Author)

I would like to thank the authors for their careful revision of the manuscript and their detailed responses to the comments and suggestions for improvement. After thorough review, I can confirm that all of these points have been well addressed and the corresponding corrections have been incorporated into the revised version. This represents a significant improvement, as additional information has been added, comprehensibility and reproducibility have been enhanced, the value and limitations of the study have been clearly stated, the structure has been refined, and repetitions have been removed.

I therefore recommend accepting the manuscript for publication and look forward to further studies on the promising approach of miniaturised flextensional ultrasonic transducers for bone surgery.

Responses to Reviewers' Comments

We thank the reviewers for their comments and recommendations. We address their comments in our responses below and have highlighted (in yellow) consequent changes made in the manuscript.

Reviewer #1:

Overall Evaluation:

This paper presents an innovative design for a miniature ultrasonic surgery device based on a flextensional configuration, in which a pre-stressed stack of piezoelectric rings serves as the ultrasonic vibration source. The prototypes demonstrate the ability to sustain high excitation levels for extended periods without showing signs of performance degradation. Although the radial dimension of the proposed design remains somewhat larger than the typical outer diameter (10–12 mm) of existing robotic instruments, this work offers a promising alternative configuration for the miniaturization of surgical ultrasonic devices.

Major Concerns:

1) This study motivates the design by stating “Ultrasonic bone scalpels are known to offer benefits” as well as “all current commercial devices are too large to be integrated with the flexible endo-wrist of a surgical robot”. However, the authors did not really propose a design with the wrist. And it is not clear in what clinical scenario, a bone scalpel with an endo-wrist is needed.

Answer:

We thank the reviewer for this comment. The aim of this paper is to investigate the design constraints of a miniature ultrasonic device for robot-assisted minimally invasive surgery, rather than to demonstrate full integration with an articulated endo-wrist. Endo-wrist has been therefore deliberately excluded to maintain a focused scope on the miniature device itself. The threaded holes at the back mass are intended for future interfacing with an endo-wrist and will be utilised in the follow-on validation studies.

We agree that the clinical motivation requires clearer articulation. The envisioned applications include, for example, spinal procedures such as osteotomies or lumbar decompressions, where enhanced dexterity and guided device manipulation are beneficial in anatomically constrained spaces.

We have revised the manuscript to clarify these scenarios and explicitly explain that endo-wrist integration is planned for future work and beyond the scope of this paper. The revision is added in the last paragraph of the introduction section.

2) The proposed miniature ultrasonic surgical device has a base size of 10 mm × 16 mm. It should be clarified whether this overall size is compatible with the anatomical constraints typically encountered in bone surgery or in endoscopic surgery.

Answer:

The 10 mm x 16 mm base dimension of the miniature device is compatible with clinically used tubular retractors for microdiscectomy, which typically have diameters of 16 – 18 mm. However, this dimension is not compatible with standard endoscopic working sleeves, which generally have diameters of 6 – 7 mm.

This clarification has been added to the revised manuscript, together with a reference.

3) The authors presented the design without any design optimization. Structure parameters may be optimized.

Answer:

The geometry of the metal frame was optimised with respect to apex height and thickness (see Fig. 1), to achieve a balanced trade-off between resonance frequency, structural stress, and displacement amplification. These two dimensions are known to most significantly affect frequency, displacement and stress. While further optimisation is possible, such as varying the number of PZT rings or adopting a curved frame profile, the design shown in this study was deliberately selected to ensure manufacturability within the dimensional tolerance achievable in our university mechanical workshop.

The rationale has been clarified in the revised manuscript, at the end of the first paragraph in 'miniature ultrasonic surgical device configuration and characterisation' section.

Fig. 1 Structural optimisation of the parameters of the metal frame.

4) A comparative analysis between the proposed flextensional ultrasonic device and a longitudinal bolted Langevin transducer (BLT) of similar base dimensions is certainly welcome and will be highly appreciated to differentiate the true values of the two designs.

Answer:

We agree with this suggestion. In our next stage work, we will conduct a systematic comparison of tissue cutting performance in specific surgical mock-ups between the miniature ultrasonic bone scalpel shown in this paper and a conventional BLT-based commercial ultrasonic surgical device, operated at comparable resonance frequencies and blade displacement amplitudes (such as Misonix BoneScalpel or Stryker Sonopet).

We are currently arranging the loan or purchase of a commercial device to support this study.

5) The issue of sealing design can be addressed or at least discussed to ensure the electrical components are protected from liquids in clinical environments. Typically, an outer casing is used to provide sealing, and it should contact the device at a zero-displacement plane (vibration node) to minimize attenuation of output displacement. The current configuration appears to have only one

such node at the base, which is intended for connection to a surgical robot. How adequate sealing can be achieved without significantly compromising displacement output requires further consideration and justification.

Answer:

We thank the reviewer for raising the important issue of sealing in liquid clinical environment. In this study, no liquid cooling was implemented and all *in vitro* bone cutting experiments were performed under dry conditions. We acknowledge that effective sealing is essential for future *in vivo* bone cutting tests, and potential solutions include epoxy encapsulation and O-rings.

This consideration has been added to the revised manuscript before 'Electrical impedance matching circuit' section in the 'Results and discussion' section.

Reviewer #2:

Congratulations to authors on a paper on an innovative miniature ultrasonic surgical device based on flextensional configuration.

The concept would benefit integration of Ultrasonic scalpel to robotics & will be very useful in spine surgery.

Answer:

We thank the reviewer for the positive feedback.

Reviewer #3:

It is recognized that authors well-conceived, executed, and documented effort to realise a miniature flextensional ultrasonic surgical device using a pre-stressed PZT ring stack. The manuscript presents a clear design rationale, validates mode shapes via FEA and LDV, investigates impedance matching both at low and high drive levels, and demonstrates promising in-vitro bone cutting results with a Kuka robot. The topic is timely and relevant to robotic minimally invasive surgery. I recommend minor revisions to address several clarifications, figure labelling, and experimental reporting issues listed below.

1) The introduction lacks a clear chapter heading. Please add a heading (e.g., "Introduction") and ensure the introductory paragraphs explicitly state the paper's objectives and the novelty relative to BLT and other flextensional approaches.

Answer:

We note that articles in *Communications Engineering* often present the first section without an explicit 'Introduction' heading. For clarity, we have retained this heading and will confirm the journal's preference with the editorial office.

The objectives and novelty of this work have been explicitly added in various paragraphs in the introduction section.

2) The "cut-through" in the frame is referred to in the text as critical for the mode-shape and for preventing bending under pre-stress, but it is not explicitly labelled in Fig. 1. Please mark the cut-through on the image.

Answer:

"Cut-through" has been annotated in Fig. 1(a) in the revised manuscript.

3) The manuscript reports lateral (F_x) and penetration (F_z) forces measured with force sensors. Please explicitly state how the static offset due to the mass of the device/fixture mounted on the sensor was accounted for (tare/calibration). Provide details: was a zeroing performed before each test? Was any drift observed during runs (and how was it compensated)?

Answer:

We thank the reviewer for this comment. Force offsets in both lateral and penetration measurements were reset to zero before each cutting experiment. Minor drift, likely due to ambient temperature changes or charge leakage, was corrected by subtracting a linearly baseline fitted between the initial and final force readings.

This clarification has been added to the 'Bone cutting test rig' section in the 'Method' section of the revised manuscript.

4) A sentence in Section 4 states: "At the higher penetration rate, the significant force reduction is lost." This is ambiguous. Please replace with an explicit quantitative statement: e.g., "At $V_z = 10 \mu\text{m/s}$, the significant force reduction is lost."

Answer:

We apologise for the inaccurate description and have clarified it in the revised manuscript by adding the value ' $10 \mu\text{m/s}$ '.

5) In Fig. 8(a), some F_z traces fall below zero. Please clarify the sign convention used for F_z (is negative tension or upward pull?) and explain the physical reason for negative readings.

Answer:

As indicated in Fig. 7 in the paper, positive F_z represents downward penetration and negative F_z upward pull. The raw X, Y, and Z force signals were noisy, so a 'smoothdata' function in Matlab was used for data processing. For $V_L = 4\text{ mm/s}$, $\Delta z = 40\ \mu\text{m}$, and $D = 1.5\ \text{mm}$, F_z ranged from $-0.1\ \text{N}$ to $0.1\ \text{N}$, which is negligible.

Despite resetting readings to $0\ \text{N}$ before each cutting experiment, the high sensitivity loadcell occasionally registered very minor negative values during Kuka robot deployment of the device, the origin of the minor negative force reading in F_z could not be determined.

6) For the case $V = 4\ \text{mm/s}$, $\Delta z = 40\ \mu\text{m}$, $D = 1.5\ \text{mm}$ the blade temperature shows a sharp rise followed by a decrease. Please explain this behaviour: change in contact area as more serrations engage, or thermal camera field-of-view/ emissivity artefacts. Provide corroborating traces (impedance, power, force) aligned with the temperature trace to support your explanation. If emissivity changes (e.g., bone debris) could affect the reading, mention this limitation.

Answer:

We thank the reviewer for this point. For the $V = 4\ \text{mm/s}$, $\Delta z = 40\ \mu\text{m}$, $D = 1.5\ \text{mm}$ robotic cutting experiment, the impedance, power, and force data does not correspond to the sudden drop in blade temperature. The most likely explanation is an instantaneous change in emissivity of the titanium blade ($\epsilon = 0.4$) caused by bone debris being ejected from the cutting site. This case represents the most 'aggressive' cutting scenario tested, producing substantially more bone debris than the $V = 2\ \text{mm/s}$, $\Delta z = 20\ \mu\text{m}$, $D = 1\ \text{mm}$, and $V = 3\ \text{mm/s}$, $\Delta z = 20\ \mu\text{m}$, $D = 1.5\ \text{mm}$ cases.

This limitation regarding bone debris affecting temperature measurement has been clarified in the revised manuscript in the 'Device integration with a Kuka robot' section in the 'Results and discussion' section.

7) The parametric matching study is thorough, but the paper should more clearly state whether the chosen matching values ($L_m = 20\ \text{mH}$, $C_m = 270\ \text{pF}$) were final for all in-vitro experiments and whether matching was re-tuned when the device encountered loading variations. Please describe the resonance-tracking controller gains/settings and how fast it can recover from detuning during cutting.

Answer:

The optimal matching parameters identified from the dynamic excitation scenario ($L_m = 20\ \text{mH}$ and $C_m = 270\ \text{pF}$) were maintained for all *in vitro* bone cutting experiments, despite change in loading and increased piezoelectric losses at high excitation.

In terms of resonance maintenance using the Piezodrive control system, it employed a constant voltage control strategy in this study to effectively track the series resonance (minimal impedance) by locking the phase between voltage and current (estimated by a phase detector) to a 0° reference. The feedback gain for phase control was set to 0.01, which was determined experimentally to achieve stable cutting performance, enabling recovery from detuning within 100 milliseconds. A detailed description of the Piezodrive resonance tracking system is provided in the following link:

<https://www.piezodrive.com/wp-content/uploads/2024/08/PDUS210-V5-Manual-R8.pdf?srsId=AfmBOors28jrAEUHwP5KIN7yVEHKMWUmuH120F3PPFDaUrGJv5dsgB0Y>

These clarifications have been added to the revised manuscript, in the last paragraph of 'Impedance matching at low and high excitation levels' section in the 'Results and discussion' section.

8) I found several small typographical issues (e.g., “noticable” → “noticeable”; “Uder” → “Under” in the test-rig results section). Please perform a careful proofread before final submission.

Answer:

We apologise for these errors, which have been corrected, and we have carefully reviewed the revised manuscript to ensure there are no grammatical mistakes.

Reviewer #4:

This article presents a novel miniaturized design of a flexible ultrasonic transducer for bone cutting. The transducer exhibits promising electromechanical behavior and achieves displacement amplitudes of up to 60 μm (pk-pk). In addition, in vitro cutting tests were performed to demonstrate its capability to cut porcine bone. The miniaturization of ultrasonic transducers is an important topic in order to integrate this technology into robot-assisted surgery and thereby perform high-precision and tissue-preserving procedures.

Overall, the manuscript is well written, and the results are novel and significant for scientists in this specific field. However, the data and, in particular, the sample size are limited, which affects the validity of the conclusion. I therefore recommend revising the manuscript to address the following key concerns. Further minor comments and recommendations can be found in the attached document.

1) The article is about a miniature ultrasonic surgical device based on a flextensional configuration with a pre-stressed PZT stack (as per the title). In addition, the article specifically describes two test modalities with two different test setups. It is a valid approach to test the feasibility and functionality of a surgical instrument in the target application (bone cutting). However, it would be beneficial to also have a commercially available device or a traditional Langevin transducer as a comparison in the same test environment to clearly demonstrate the added value of the novel concept. Although it is generally demonstrated that the present concept has potential, it is not entirely clear which advantages can be attributed to the novel transducer concept, the impedance matching and the corresponding resonance tracking, or the shape of the cutting blade. I therefore recommend supplementing the discussion of these points by taking relevant literature into account.

Answer:

We thank the reviewer for these comments. For response, please see the following section (below), titled 'minor comments and recommendation', referred to as **comment (2)** on **Page 11**.

2) The section "Results and Discussion" contains a lot of information on the methodology and the materials used (e.g. test rigs). These are described in more detail in the section "Methods". To avoid redundancy, I recommend describing all test setups and characterization methods in detail only in the chapter on methods.

Answer:

We thank the reviewer for this comment. For response, please see the following section (below), titled 'minor comments and recommendation', referred to as **comment (3)** on **page 2**.

3) Is the proposed flextensional design suitable for use in minimally invasive robot-assisted surgery? The introduction mentions that traditional transducers (BLT) are not suitable for this purpose due to their size. However, there is also the option of using an extension or a long cutting blade. What would be the limitations in this case? Furthermore, does the proposed flexural transducer fit through a trocar? This could be explored in more detail in the discussion.

Answer:

We thank the reviewer for this comment. The question regarding 'However, there is also the option of using an extension or a long cutting blade. What would be the limitations in this case?', for response, please see the following section (below), titled 'minor comments and recommendation', referred to as **comment (3)** on **page 1**.

The proposed miniature ultrasonic bone scalpel (10 mm × 16 mm at the base × 45 mm length) is compatible with tubular retractors used in microdiscectomy (16 – 18 mm diameter), the intended application for minimally invasive spinal surgery, though it is slightly larger than the largest standard trocar (15 mm). This clarification has been added to the revised manuscript, together with a reference. Further miniaturisation, for example by reducing the number and diameter of the piezoelectric rings, will be investigated in future work.

4) The description of the cutting blade design requires further explanation: What are the design considerations for the cutting blade? It appears to be quite different from commercially available ultrasonic osteotomy blades. In addition, why was the blade not designed as a separate part with, for example, a thread? For the target application, the entire transducer would be a disposable product. This can also be mentioned as a limitation in the discussion.

Answer:

We thank the reviewer for these comments. For response, please see the following section (below), titled ‘minor comments and recommendation’, referred to as **comment (1)** and **comment (7)** on **page 2**.

5) “Surgical robot”: Is the Kuka robot used a surgical robot? According to the Kuka website, the LBR iiwa 14 R820 model is a collaborative industrial robot. However, Kuka also offers surgical robots, but in this case, these would be the LBR Med 7 R800 or LBR Med 14 R820 models. It would be advisable to rename the “surgical robot” or clearly describe that a test setup with an industrial robot is being used and how it differs from the version used for surgical procedures.

Answer:

We thank the reviewer for this comment, which has been address in the following minor comments and recommendations, in **comment (2)** on **page 2**.

6) The description of the test rig and the cutting process is clear and comprehensible. However, it is not entirely clear why this type of sawing motion was chosen for the test, as this is rarely the case when using ultrasonic osteotomy devices. Could a more detailed explanation be provided in this regard? Furthermore, such a sawing motion and penetration rate are difficult to implement in robot-assisted surgery and are likely to be too slow and imprecise for clinical requirements.

Answer:

We thank the reviewer for these comments, which have been address in the following minor comments and recommendations, in **comment (3)** on **page 6**.

7) Results: Is there an explanation for the fact that at 10 $\mu\text{m/s}$, the forces at 30 μm are even higher than with the non-activated transducer, and then at 15 $\mu\text{m/s}$ they are lower compared to 0 μm and 60 μm or lower cutting speeds? One possible reason, as also mentioned later in the corresponding paragraph, could be that the material tested is biological samples, which by nature exhibit a high variation in mechanical properties (e.g. thickness of cortical bone). There are also synthetic bone models that may allow for more standardized testing of device performance. Furthermore, were the measurements repeated? If so, what is the sample size and what is the standard deviation (statistical analysis)? If not, further measurements on biological or synthetic samples may be necessary in order to make a proper assessment.

Answer:

We thank the reviewer for these comments, which have been address in the following minor comments and recommendations, in **comment (2)** and **comment (3)** on **page 8**.

Minor Comments and Recommendations:

Page 1:

1) Rather “shorter”.

Answer:

The word has been updated in the revised manuscript.

2) Are there other concepts for miniaturizing surgical ultrasonic transducers, such as planar designs, folded structures, or structures with high compliance?

Answer:

We thank the reviewer for suggesting additional configurations (planar and folded designs) for the miniaturisation of ultrasonic surgical devices. We agree that other design options with strong miniaturisation potential exist, although they were beyond the scope of the present study.

Explanation with relevant literature has been added to the ‘Introduction’ section in the revised manuscript, outlining the advantages and limitations of the planar and folded designs. We also note that multiple terms are used in the literature to describe these configurations (e.g. planar and folded flextensional transducers, planar ultrasonic scalpels, and twice planar folded structures used as the front mass of BLT transducers), which are now clarified in the manuscript.

3) What about an extension? Ultrasonic osteotomy devices with an extension for minimally invasive procedures through a trocar are already available.

Answer:

We thank the reviewer for this suggestion. We agree that ultrasonic osteotomy devices with extensions for minimally invasive procedures through a trocar are already available. However, such devices are typically driven by BLT transducers coupled to long, slender shafts (approximately 5mm in diameter and 400 mm in length). As discussed in the ‘Introduction’ section, these long and rigid instruments are prone to a pronounced ‘fulcrum’ effect at the point of insertion into the human body, which can amplify hand tremor and reduce controllability. In addition, the extended straight waveguide with an attached surgical tip is susceptible to flexural vibrations, leading to undesirable heat generation during tissue cutting.

In contrast, the miniature flextensional transducer proposed in this paper is intended to be integrated with a robot endo-wrist, enabling the device to benefit from the seven degrees-of-freedom, offering significantly increased dexterity and access to difficult-to-access surgical sites. This approach is fundamentally incompatible with long, rigid extensions, which motivates the proposed design.

4) What is the maximum size/diameter?

Answer:

The standard largest trocar used in minimally invasive surgery is 15 mm, typically employed for very large instruments or specimen retrieval.

This number, together with a reference, has been added to the revised manuscript in the 'Introduction' section.

5) Does this really apply to all classes? Are there possibly also configurations of classes I to VII that do not use any bonding material? This claim sounds speculative, and I would suggest that it be worded more precisely.

Furthermore, the reference provided only describes a cymbal transducer (class V). I recommend providing another reference that describes all classes.

E.g.:

Pyun, J. Y., Kim, Y. H., & Park, K. K. (2023). Design of Piezoelectric Acoustic Transducers for Underwater Applications. *Sensors*, 23(4), 1821. <https://doi.org/10.3390/s23041821>

Answer:

We thank the reviewers for suggesting an additional reference on flextensional transducer classes. This reference has been added to the revised manuscript in the 'Introduction' section, along with clarification of bonding-related limitations in existing flextensional designs with other relevant reference papers.

Page 2:

1) Why wasn't the blade designed as a separate part with, for example, a thread? For the target application, the entire transducer would be a disposable product. This can also be mentioned as a limitation in the discussion.

Answer:

Fig. 2 The miniature flextensional transducer attached with Mectron US1 and US2 surgical tips by means of a threaded stud.

We thank the reviewer for suggesting a disposable cutting blade as a separate component of the flextensional transducer. This concept was evaluated using Mectron US1 and US2 dental tips (see the following Fig. 2), and initial handheld bone cutting tests demonstrated outstanding performance. However, a threaded disposable tip introduces several challenges, including difficulty in ensuring consistent blade alignment, which can result in unbalanced dynamic behaviour during excitation. Over-tightening may induce microcracks in the thin metal shell that propagate under high voltage excitation, while under-tightening risks blade loosening and increased mechanical damping, leading to excessive heat generation. Although an optimal tightening torque can be calculated, achieving reliable assembly in practice remain non-trivial.

The prototype presented in this paper therefore employs a monolithic metal shell with an integrated blade, eliminating alignment issues and interface induced damping. The limitations of this approach and future work toward a clinically viable disposable solution are now discussed in the 'Miniature ultrasonic surgical device configuration and characterisation' section in the 'Results and discussion' section.

2) Is the Kuka robot a surgical robot? According to the Kuka website, the LBR iiwa 14 R820 model is a collaborative industrial robot. However, Kuka also offers surgical robots, but in this case, these would be the LBR Med 7 R800 or LBR Med 14 R820 models. It would be advisable to rename the "surgical robot" or clearly describe that a test setup with an industrial robot is being used and how it differs from the version used for surgical procedures.

Answer:

The Kuka robot LBR iiwa 14 R820 use in this study is an industrial robot available at the STORM lab, University of Leeds, during the bone cutting experiments. Its repeatability ($\pm 0.1 - 0.15$ mm) is comparable to that of the medical robots LBR Med 7 R800 and LBR Med 14 R820. The robotic arm was used solely to position the miniature device on the bone, while a higher precision Hexapod (± 0.1 μ m repeatability) performed the bone cutting.

In the revised manuscript, 'Kuka surgical robot' has been renamed to 'Kuka robot', with its usage described in the 'Methods' section with a reference.

3) The section "Results and Discussion" contains a lot of information on the methodology and the materials used (e.g., test rig). These are described in more detail in the section "Methods." To avoid redundancy, I recommend describing all test setups and characterization methods in detail only in the chapter on methods.

Answer:

We thank the reviewer for noting this redundancy. Description of the device, impedance matching circuit, test rig, and robotic platform have been moved to the 'Methods' section. Key parameters (V , L , D , and displacement amplitude) are retained in the 'Results and discussion' section, as they are essential for interpreting the cutting results.

4) I agree that this is very unusual behavior. Are there any references where such capacitive behavior has also been observed at series resonance?

Answer:

We thank the reviewer for highlighting the capacitive behaviour at the series resonance in the miniature flextensional ultrasonic device. Literature on class IV and V flextensional transducers often reports high impedance magnitude but rarely shows phase-frequency characteristics.

We attribute this behaviour to the interaction between mechanical resonance, electromechanical coupling, and the compliance of the metal shell. Unlike BLTs, which exhibit nearly purely resistive impedance, flextensional transducers transform the small strain of the piezoelectric stack into large blade displacement. This introduces a phase lag between force and velocity, producing a non-resistive impedance. While displacement is amplified, force and effective electromechanical coupling ($k_{\text{eff}} \ll k_{33}$) decreases significantly, resulting in a non-neutralised static capacitance C_0 at the series resonance.

Experimental validation on other flextensional classes is needed to confirm these mechanisms, which is beyond the scope of this study.

Explanation has been added to the revised manuscript, in the 'Miniature ultrasonic surgical device configuration and characterisation' section in the 'Results and discussion' section.

5) Fig. 1 (a) only shows the CAD. Fig 1 (c) & (d) shows the FEA results.

Answer:

Yes, Fig. 1 shows the exploded CAD of the miniature ultrasonic surgical device, with (c) the predicted FEA vibration mode and (d) its validation in EMA.

6) How was Q calculated? 3dB-method or transient response?

Answer:

Mechanical quality factor Q in this study was calculated using the 3dB method at the series resonance, based on the electrical impedance of the ultrasonic surgical device.

This explanation has been added to the revised manuscript above equation (1).

7) What are the design considerations for the cutting blade? It appears to be quite different from commercially available ultrasonic osteotomy blades.

Answer:

The scalpel-serration blade designed in this study allows both manual and automated bone cutting, supporting sawing and chiselling mechanisms. We have also developed alternative blade geometries with this transducer configuration, similar to the MXB-B1 blade of the Misonix BoneScalpel, which successfully cut bone samples. Optimisation of the transducer and blade design will be investigated in future research to further enhance cutting performance.

Explanations have been added to the revised manuscript, in the first paragraph of the 'Miniature ultrasonic surgical device configuration and characterisation' section in the 'Results and discussion' section.

8) Where exactly? At the tip of the cutting blade? Could also be indicated with an arrow in Fig. 1 d.

Answer:

Yes, the displacement amplitude was measured at the tip of the cutting blade. The term 'blade tip' has been clarified, and an annotated arrow has been added to Fig. 1 (d) in the revised manuscript.

9) How large was the set frequency increment?

Answer:

The frequency increment in the harmonic analysis was set to 5 Hz, allowing detailed observation of the change in the vibration characteristics near the series resonance.

The explanation has been added to the revised manuscript.

10) Will this fit through a trocar?

Answer:

The 10 mm x 16 mm base dimension of the miniature device can pass through clinically used tubular retractors (16 – 18 mm diameter) for minimally invasive microdiscectomy.

This explanation has been added to the revised manuscript with a reference, in the 'Miniature ultrasonic surgical device configuration and characterisation' section in the 'Results and discussion' section.

Page 3:

1) A reference should be added for this statement.

Answer:

We thank the reviewer for this suggestion. Two reference papers have been added to the revised manuscript.

2) Was operation at the anti- (parallel) resonant frequency considered? What are the arguments for/against this? There are commercial ultrasonic osteotomy systems that use fa drive strategy.

Answer:

We thank the reviewer for highlighting the interesting concept of driving the miniature device at anti-resonance. We are not aware of the specific drive strategies or impedance matching mechanisms used in commercial ultrasonic osteotomy devices.

Parallel resonance features high impedance, making the system less responsive to rapidly varying mechanical loads during bone cutting, whereas series resonance, having much lower impedance, provides more sensitive response. Although the PDUS-210 resonance tracker (Piezodrive, Australia) supports parallel resonance driving, we chose series resonance driving with an LC impedance matching circuit. Achieve 60 μm blade amplitude requires 90 V and 3.6 W; parallel resonance would require substantially higher voltage, risking system overload. Moreover, the impedance of the miniature device at parallel resonance shows multiple noisy peaks, likely due to the current sensor's limited ability to detect very low currents, which could compromise the resonance tracking performance.

Nevertheless, investigation of parallel resonance driving remains worthwhile, as it may improve dynamic stability and current limiting under varying mechanical loads. Future work will compare series and parallel resonance tracking strategies to assess their suitability for the miniature flextensional ultrasonic surgical device.

Page 4:

1) add a point "."

Answer:

A point "." has been added to the caption of Fig. 2 in revised manuscript.

2) It is difficult to understand why the highest Keff value "a measure of the surgical device's conversion efficiency from electrical energy to mechanical vibration" results in the lowest Q value and the smallest displacement (Fig. 3 b). Is there an explanation for this?

Answer:

We thank the reviewer for this observation. The following Fig. 3 illustrates the effect of the LC matching circuit on device impedance. We show the changes in impedance measurements when $C_m =$

300 pF while increasing L_m from 0 to 30 mH, and when $L_m = 30$ mH while increasing C_m from 0 to 300 pF.

Increasing L_m and C_m shifts the series resonant frequency f_r to a lower value, while the parallel resonant frequency f_a remains nearly constant. The impedance peaks broaden, indicating reduced mechanical Q , and the impedance magnitude decreases from ~ 2 k Ω (unmatched device) to < 100 Ω . This, combined with the LC circuit's losses and mismatch with the PDUS-210 resonance tracker (~ 400 Ω optimal), results in the lowest displacement amplitude shown in Fig. 3 (b) in the paper, even lower than that of the unmatched device.

With the LC circuit, the mechanical resonator behaves as a frequency-dependent electrical load. Increasing L_m and C_m enhances electrical energy extractions and losses, which presents as the increase in electromechanical coupling K_{eff} associated with added mechanical damping, reflected in the reduced mechanical Q of the miniature device.

Fig. 3 Impedance measurement when (a) increasing L_m from 0 to 30 mH with $C_m = 300$ pF, and (b) increasing C_m from 0 to 300 pF with $L_m = 30$ mH

3) The word “static” can be misleading here. Does it refer to operation at DC voltage or rather to a very low deflection due to low power? I suggest omitting “static” and only distinguishing between low and high power.

Answer:

We apologise for the misleading use of the term ‘static’ in the original manuscript. Here it refers to a low power excitation of the miniature device using an electrical impedance analyser, rather than a DC voltage.

In the revised manuscript, the term ‘static’ has been replaced with ‘low-level excitation’, and the term ‘dynamic’ has been replaced with ‘high-level excitation’ to more accurately describe the testing conditions.

4) You might consider adding the control (resonance tracking) to Fig. 2.

Answer:

We thank the reviewer for this suggestion. The resonance tracking system used to drive the miniature ultrasonic surgical device is already shown in Fig. 2 in the paper, as the Piezodrive Ultrasonic Driver above V_E .

Fig. 2 primarily illustrates the electrical impedance matching framework employed in this study, namely the *LC* configuration. The miniature ultrasonic surgical device is additionally represented using the Butterworth Van-Dyke equivalent circuit.

Page 6:

1) This argument is not entirely comprehensible, as the both the MSA100 and OFV 303 can measure up to 10 m/s, which at a frequency of 23896 Hz would correspond to a peak-to-peak displacement of 133 μm .

Answer:

We thank the reviewer for highlighting measurement capabilities of the laser Doppler vibrometer, for both (MSA-100 and OFV-303). The 60 μm peak-to-peak amplitude reported reflects a limitation of the measurement set-up, not the LDV's range at ~ 24 kHz. To measure the velocity, we attached a small laser reflective sticker to the blade tip, which frequently detached at higher frequency and displacement, demonstrating the practical difficulty of maintaining it during high voltage excitation. Therefore, 60 μm was the maximum amplitude achievable while keeping the laser focused on the blade tip.

For future work, we plan to explore alternative displacement measurement methods for the miniature device, such as ultra-high speed imaging.

2) Has the emissivity been calibrated? The cutting side (bone or cutting blade) may have a different emissivity than the PZT stack.

Answer:

Yes, the emissivity for temperature measurement at the cutting-site (blade) was set to 0.4 in the TIM 160 thermal camera, reflecting its relatively rough machined surface. For the PZT stack, the outer circular face of the PIC181 ring (dark-grey ceramic surface, nearly black) has an emissivity of 0.9.

3) The description of the setup and the cutting process is clear and comprehensible. However, it is not entirely clear why this type of sawing motion was chosen for the test, as this is rarely the case when using ultrasonic osteotomy devices. Could a more detailed explanation be provided in this regard? Furthermore, such a sawing motion and penetration rate are difficult to implement in robot-assisted surgery and are likely to be too slow and imprecise for clinical requirements.

Answer:

We thank the reviewer for this comment. The reciprocal sawing motion, combined with controlled device penetration, was intentionally chosen as a simplified, highly repeatable experimental protocol to ensure stable and coupled blade and bone interaction, enabling systematic comparison of cutting force, temperature, and the electrical characteristics of the device.

We acknowledge that this motion does not reflect clinical use of ultrasonic osteotomy devices, and may be difficult to implement in robot-assisted surgery; it serves as a laboratory characterisation method. As noted in the last paragraph of the 'Results and discussion' section, we have mentioned that 'Additionally, integrating haptic mechanisms with force feedback strategies could improve the motion control of the robot, enabling more precise bone cutting along pre-defined pathways.' In future work, we plan to implement slow axial advancement of the device, which is assisted with lateral guidance with the robot, instead of continuous sawing motion.

This explanation has been added to the revised manuscript, in the 'Ultrasonic bone cutting experiments' section in the 'Results and discussion' section.

Page 8:

1) Can an outlook be provided as to whether the resonance tracking system, the transducer prototype or the cutting blade requires optimization?

Answer:

We thank the reviewer for the comment. Optimisation of the structural parameters of the miniature ultrasonic surgical device and the design of the cutting blade has been added to the 'Miniature ultrasonic surgical device configuration and characterisation' section in the 'Results and discussion' section. Additionally, an explanation of the resonance tracking system (Piezodrive) has been added to the last paragraph of the 'Impedance matching at low and high excitation levels' section in the 'Results and discussion' section of the revised manuscript.

2) Is there an explanation for the fact that at 10 $\mu\text{m/s}$, the forces at 30 μm are even higher than with the non-activated transducer, and then at 15 $\mu\text{m/s}$ they are lower compared to 0 μm and 60 μm or lower cutting speeds?

One possible reason, as also mentioned later in this paragraph, could be that the material tested is biological samples, which by nature exhibit a high variation in mechanical properties (e.g., thickness of cortical bone). There are also synthetic bone models that may allow for more standardized testing of device performance.

See for example: O. F. Okaya, J. Burger, and M. Hofmann, "Additively Manufactured Ultrasonic Osteotomy Inserts for Improved Temperature Control and Tissue Preservation," 2024 52nd Annual Ultrasonic Industry Association Symposium (UIA), Dublin, Ireland, 2024, pp. 1-5, doi: 10.23919/UIA60812.2024.10716027.

Answer:

We thank the reviewer for highlighting the choice of porcine bone material in this study, and for referencing the work using synthetic bone material in "Additively Manufactured Ultrasonic Osteotomy Inserts for Improved Temperature Control and Tissue Preservation".

In our previous study, 'Can Mn:PIN-PMN-PT piezocrystal replace hard piezoceramic in power ultrasonic devices?' (Ultrasonics), we compared BLT surgical devices with hard PZT and single crystal materials using both synthetic bone (sawbone) and chicken femur for a comprehensive assessment.

However, in this present study, we selected the more clinically relevant porcine bone, which has properties closer to those of human bone, for cutting experiments on the test rig and with the Kuka robot. Variations in bone geometry and regional hardness likely explain why the cutting force F_x at 10 $\mu\text{m/s}$ penetration rate and 30 μm blade amplitude has resulted in a higher value than that of non-ultrasonic cutting. Additionally, deeper device penetration may engage more bone surface area, increasing contact compared to other tests. These factors illustrate the inherent variability of animal tissue versus uniform synthetic materials.

Further explanation has been added to the revised manuscript, in the 'Ultrasonic bone cutting experiments' section in the 'Results and discussion' section.

3) Were the measurements repeated? If so, what is the sample size and what is the standard deviation? If not, further measurements on biological or synthetic samples may be necessary in order to make a proper assessment.

Answer:

We thank the review for raising this question. The cutting experiments on the test rig were not repeated, primarily to avoid wear of the teeth, especially for non-ultrasonic cutting (0 μm amplitude). We also aimed to use clinically relevant biological samples (porcine ribs) rather than synthetic sawbone.

Two newly fabricated miniature ultrasonic surgical devices were used for the test rig experiments (9 cuts, see Fig. 5 in the paper) and the Kuka robot experiments (3 cuts, see Fig. 8 in the paper), minimising degradation of the blade teeth. Additionally, fresh porcine ribs exhibit variations in geometry and mechanical properties across regions, which likely contributed to the unusual force observation in Fig. 5 in the paper.

We acknowledge the value of testing the miniature device under various parameters using synthetic samples with consistent properties to more comprehensively evaluate the cutting performance, and plan to conduct such cutting experiments in future work.

Explanations have been added to 'Ultrasonic bone cutting experiments' section in the 'Results and discussion' section of the revised manuscript.

4) Why was an irrigation system, which is typically used for flushing and cooling during ultrasonic osteotomy, not implemented in the test bench?

Answer:

We thank the reviewer for highlighting the important issue of sealing in liquid clinical environment. In this study, the proposed design did not incorporate a liquid cooling system, and all *in vitro* bone cutting tests were performed in a dry environment. We intentionally avoid integrating flushing or cooling, as the presence of liquid would affect temperature measurements at the cutting site, which is a key parameter for understanding the cutting speed and displacement amplitude required to maintain effective blade and bone interaction (vibro-impact response). We acknowledge, however, that proper sealing of the miniature device will be essential for future *in vivo* bone cutting experiments.

This discussion has been added to the revised manuscript, in the last paragraph in the 'Miniature ultrasonic surgical device configuration and characterisation' section in the 'Results and discussion' section.

Page 11:

1) ", USA"? To keep the naming consistent with the "Results" section. In general, the manuscript should be checked for consistency in naming.

Answer:

We apologise for the omission of the countries of origin of the equipment in the original manuscript.

The countries of the manufacturers' headquarters have now been added to the revised manuscript. The duplicated description of the instruments in the 'Results and discussion' section has been moved to the 'Methods' section.

2) The article is about a miniature ultrasonic surgical device based on a flextensional configuration with a pre-stressed PZT stack (as per the title). In addition, the article specifically describes two test modalities with two different test setups. It is a valid approach to test the feasibility and functionality of a surgical instrument in the target application (bone cutting). However, it would be beneficial to also have a commercially available device or a traditional Langevin transducer as a comparison in the same test environment to clearly demonstrate the added value of the novel concept. Although it is generally demonstrated that the present concept has potential, it is not entirely clear which advantages can be attributed to the novel transducer concept, the impedance matching and the corresponding resonance tracking, or the shape of the cutting blade. I therefore recommend supplementing the discussion of these points by taking relevant literature into account.

Answer:

We thank the reviewer for suggesting a comparison with a commercially available device. We are currently in the process of obtaining a commercial surgical device (e.g. Misonix BoneScalpel or Stryker Sonopet with similar resonance frequency and blade amplitude) in preparation for the next stage comparison study.

In the revised manuscript, the final paragraph of the 'Results and discussion' section has been expanded to more clearly summarise the contributions of the proposed transducer concept, impedance matching and resonance tracking, and blade geometry.

Page 12:

1) A more detailed explanation is needed as to why a hexapod is used for the cutting movement and why not just the Kuka robot.

Answer:

This explanation has been added to the 'Robotic bone cutting platform' section in the 'Methods' section of the revised manuscript.

A miniature ultrasonic surgical device based on a flextensional configuration with a pre-stressed PZT stack

Xuan Li^{1,2}, Dominic Jones³, Pietro Valdastrì³ and Margaret Lucas¹

¹Centre for Medical & Industrial Ultrasonics, James Watt School of Engineering, University of Glasgow, UK

²Department of Mechanical Engineering, School of Engineering, University of Southampton, UK

³STORM Lab UK, School of Electronic & Electrical Engineering, University of Leeds, UK

Abstract — Ultrasonic bone scalpels are known to offer benefits of low cutting force, high precision, low microdamage around the cut site, and tissue selectivity in surgical procedures. However, all current commercial devices are too large to be integrated with the flexible endo-wrist of a surgical robot and therefore there is a significant gap for innovation in miniature devices. Ultrasonic bone scalpels in use in clinical settings are all based on a bolted Langevin transducer (BLT), which consists of a pre-stressed piezoceramic ring stack, two end masses, and a cutting blade. The BLT-based device must operate in resonance to achieve sufficient displacement amplitude at the surgical tip to cut through bone, and this dictates its size. Flextensional transducers have emerged as an alternative, but these transducers generally contain a low volume of piezoelectric driving material, and hence cannot excite the required displacement amplitude, and their reliance on adhesive bonds in their fabrication means they fail at the excitation levels required for a bone surgery device. We present a novel flextensional configuration that forms an ultrasonic surgical device, where the vibration amplifying metal caps are excited by a pre-stressed piezoelectric stack. *In vitro* ultrasonic bone cutting tests facilitated with a Kuka surgical robot are performed for a range of cutting speeds and penetration rates. The results demonstrate effective integration with a surgical robot and that bone cutting can be achieved with an extremely low cutting force (<1 N) and high precision (width of the bone cut presents under 6% deviation from the thickness of the blade).

The benefits of ultrasonic bone surgery devices are well established and include high precision, low cutting force, reduced trauma, less collateral damage, and sparing of other tissue structures [1], [2], [3], [4], [5]. Ultrasonic bone surgery devices operate at a low ultrasonic frequency, usually between 20 and 35 kHz, causing bone fragmentation due to high frequency impacting in the cut depth direction for scalpel-like blades or superposition of high frequency oscillations on the sawing action of the serrated blades. Ultrasonic bone surgery was originally invented for maxillofacial, periodontal and endodontic surgeries [6], [7], but applications have extended to many other surgical procedures, including skull base and spinal surgeries [3], [8], [9], [10], [11].

Conventional minimally invasive surgeries present practical limitations. For example, laparoscopic/endoscopic surgeries, through single or multiple incisions, use long and rigid surgical instruments that often suffer from the ‘fulcrum’ effect caused by the point of insertion into the body [12], [13], [14], [15]. This fulcrum acts as a point of rotation which inverts the surgeon’s movements and can amplify hand tremor, making the instrument more difficult to use. Surgical robots have been introduced to clinical practice to mitigate these limitations, enhance dexterity, improve stability, and increase motion

accuracy [12]. This technology is known as robotically assisted minimally invasive surgery (RAMIS) and originates from laparoscopic surgical procedures facilitated by the da Vinci surgical robot [16], [17], [18], [19].

Incorporating ultrasonic surgical devices into RAMIS creates additional challenges. A long straight waveguide attached to the bolted Langevin transducer (BLT) is required to excite vibration of the distal surgical tip, but these can suffer from flexural vibrations, and hence heating [20], [21]. Also, this configuration of ultrasonic device is restricted in its degrees of freedom at the surgical site. The smaller, hand-held, ultrasonic devices cannot integrate with surgical robots; they do not fit through the trocar, which is typically 10 to 12 mm diameter [13], [14], [15], [22], [23], [24], [25], and they cannot connect to the robot arm. An alternative, much more dexterous, method for delivering surgical instruments is through integration with the robot’s endo-wrist, which can enable seven degrees-of-freedom. However, to connect with an endo-wrist and comply with trocar size, a miniature ultrasonic surgical device is required.

Miniaturisation presents challenges due to the small volume of usable piezoelectric material, making the device less powerful than the larger commercial ultrasonic surgical devices. Driving a miniature, but conventionally configured BLT at higher power is not a solution; the result is detrimental thermal effects, significant loss in piezoelectric performance and undesirable nonlinear behaviour. Additionally, because a **1** smaller BLT is achieved by increasing the resonance frequency, a much lower vibrational displacement amplitude is excited in the cutting blade [4] and the performance needed for effective bone cutting cannot be achieved.

2 flextensional transducer is an alternative configuration for device miniaturisation. This type of transducer is commonly used in low to medium ultrasonic frequency and high-power underwater projectors, which radiate sound by the flexure of a metal shell excited by a piezoelectric plate operating in extensional vibration [26]. Lab-based prototypes of an ultrasonic surgical device based on the flextensional configuration have been reported that demonstrate cutting of bone tissue [27], [28], [29]. **3** however, the size is still too large for entry via a **4** trocar due to the need to accommodate sufficient piezoelectric material to achieve the required displacement amplitude of the cutting blade. **5** major issue with the configuration of the conventional classes of flextensional transducer is that they rely on a bonding agent (usually an epoxy resin) to secure the metal shell to the piezoelectric material and this softens and fails at the high excitation levels required [29], limiting the devices to low displacement amplitude applications.

Summary of Comments on COMMS-25-0581-T

Page: 1

Number: 1
rather "shorter"

Number: 2
Are there other concepts for miniaturizing surgical ultrasonic transducers, such as planar designs, folded structures, or structures with high compliance?

Number: 3
What about an extension? Ultrasonic osteotomy devices with an extension for minimally invasive procedures through a trocar are already available.

Number: 4
What is the maximum size/diameter?

Number: 5
Does this really apply to all classes? Are there possibly also configurations of classes I to VII that do not use any bonding material? This claim sounds speculative, and I would suggest that it be worded more precisely.

Furthermore, the reference provided only describes a cymbal transducer (class V). I recommend providing another reference that describes all classes.

E.g.:

Pyun, J. Y., Kim, Y. H., & Park, K. K. (2023). Design of Piezoelectric Acoustic Transducers for Underwater Applications. *Sensors*, 23(4), 1821. <https://doi.org/10.3390/s23041821>

Mechanical fixings along with an adhesive bond can enable higher displacement amplitudes to be reached [28], [29], but the achievable displacement is still too low, and it does not solve the problem of the size of the device being too large for integration with a surgical robot endo-wrist.

An innovative design of a miniature ultrasonic surgical device based on a flextensional configuration is presented, Fig. 1(a). The device is distinctive from the known classes of flextensional transducer [26], which employ a piezoelectric plate or disc or bar as the ultrasonic vibration source, and then transform the small extensional motion of the driving element into flexural motion of the metal shell [28], [29], [30]. The new device employs a pre-stressed stack of piezoelectric rings connected to a single-piece metal frame, which functions as the mechanical amplifier and **incorporates the surgical cutting blade**. We demonstrate that prototypes based on this design are capable of sustaining high excitation levels for a long duration in bone cutting experiments, without showing signs of deterioration in cutting performance. This is the first report of a miniature ultrasonic surgical device prototype that has been successfully integrated with a **surgical robot** for hard tissue resection.

Results and discussion

Miniature ultrasonic surgical device configuration and characterisation

The miniature ultrasonic surgical device was designed in CAD and modelled in finite element analysis (FEA) (Abaqus, Dassault Systèmes, France), Fig. 1(b). The device is comprised of a stack of lead zirconate titanate (PZT) piezoelectric rings pre-stressed in a single-piece metal frame by a threaded bar. The frame is manufactured from the titanium alloy Ti-6Al-4V and has a working side, which incorporates the surgical blade, and a connecting side, which provides a connector to the surgical robot (Fig. 1(b)). The frame therefore acts as a one-sided flextensional transducer, with an integrated serrated **cutting blade**.

Unlike the standard classes of flextensional transducer that are symmetrical and exhibit a symmetric mode of vibration (high deformation on both sides), the surgical device requires a non-vibrating connection to the surgical robot. The geometry of the connection side of the device is therefore carefully designed to meet a number of requirements: (i) it must be configured to ensure there is no vibration at the robot connection location, (ii) the mass and geometry must balance the overall vibrational mode such that the PZT stack deforms without any bending, (iii) it must not limit the vibrational amplification of the working flextensional side, (iv) it must prevent distortion of the frame when the PZT stack pre-stress is applied during fabrication, and (v) it must enable the device to maintain high dynamic stability during bone cutting. A simple frame geometry, that importantly includes a cut-through as seen in Fig. 1, enables all these requirements to be met.

The overall size of the surgical device is **10 mm × 16 mm at the base × 45 mm length**. The blade has a cutting tip of 7.75 mm length and 0.5 mm width. These dimensions created a device that can be tuned to a target resonance frequency comparable

with commercial hand-held ultrasonic bone surgery devices (usually in the 20 – 30 kHz range).

At the operating resonance frequency of the device, the frame amplifies the small axial displacement of the PZT ring stack into a large flexural deformation of the working side of the frame and hence into axial vibrational displacement of the blade. This can be seen in the FEA model result of the vibrational mode shape in Fig. 1(c) which is validated by scanning laser Doppler vibrometer (LDV) (MSA100, Polytec, Germany) measurement of the mode shape in Fig. 1(d).

The FEA predicted and experimentally measured mode shapes, which are in close agreement, also show that there is no vibration of the connector side of the transducer and no bending in the PZT stack, which positively affects the electromechanical efficiency of the device [31], [32], [33]. The predicted and measured resonance frequencies vary by 7%; f_r is 22273 Hz from the FEA model and 23872 Hz from the LDV measurement. This variation is ascribed to the dimensional tolerance of the cut-through in the metal frame, which is found to have a strong effect on frequency when pre-stress is applied to the PZT stack.

An impedance analyser (4294A, Agilent, USA) was used to measure the impedance magnitude and phase of the device at the resonance frequency and the results are shown in Fig. 1(e). The resonance frequency f_r is measured to be 23896 Hz and the impedance magnitude and phase at f_r are 1927 Ω and -40.8° , respectively. This highlights a significant challenge for the miniature surgical device; BLTs typically exhibit an impedance magnitude an order of magnitude lower and a phase of 0° . The miniature device therefore exhibits **capacitive characteristics at resonance** and requires an impedance matching circuit and a resonance tracking system.

The electromechanical coupling coefficient k_{eff} is calculated from the impedance spectrum in Fig. 1(e) using equation (1) [34], providing a measure of the surgical device's conversion efficiency from electrical energy to mechanical vibration. The **mechanical quality factor, Q** , is also calculated as an important indicator of the surgical device's potential to achieve large displacement amplitude with low losses.

$$k_{\text{eff}}^2 = \frac{f_a^2 - f_r^2}{f_a^2} \quad (1)$$

The measured coupling coefficient k_{eff} is calculated to be 0.053, which is significantly lower than a conventional BLT that typically has a value ranging from 0.2 to 0.5 [35], [36], [37]. This low k_{eff} is due to the high impedance magnitude and capacitive characteristics of the device at the resonance frequency. Despite this, the mechanical quality factor Q is almost 600, which is more representative of a BLT, which is often in the range of several hundred to a few thousand [38], [39].

The result of a harmonic analysis of the device is shown in Fig. 1(f). The displacement of the **blade tip** is measured in an upward frequency sweep through the resonance frequency using an LDV, at **increasing increments** of excitation voltage. The device exhibits a characteristic nonlinear softening with the backbone curve of the frequency response bending slightly towards the left [40] as excitation voltage is incremented from 1 V_{rms} up to 100 V_{rms} . At the highest excitation level, a displacement

Number: 1

Why wasn't the blade designed as a separate part with, for example, a thread? For the target application, the entire transducer would be a disposable product. This can also be mentioned as a limitation in the discussion.

Number: 2

Is the Kuka robot a surgical robot? According to the Kuka website, the LBR iiwa 14 R820 model is a collaborative industrial robot. However, Kuka also offers surgical robots, but in this case, these would be the LBR Med 7 R800 or LBR Med 14 R820 models. It would be advisable to rename the "surgical robot" or clearly describe that a test setup with an industrial robot is being used and how it differs from the version used for surgical procedures.

Number: 3

The section "Results and Discussion" contains a lot of information on the methodology and the materials used (e.g., test rig). These are described in more detail in the section "Methods." To avoid redundancy, I recommend describing all test setups and characterization methods in detail only in the chapter on methods.

Number: 4

I agree that this is very unusual behavior. Are there any references where such capacitive behavior has also been observed at series resonance?

Number: 5

Fig. 1 (a) only shows the CAD. Fig 1 (c) & (d) shows the FEA results.

Number: 6

How was Q calculated? 3dB-method or transient response?

Number: 7

What are the design considerations for the cutting blade? It appears to be quite different from commercially available ultrasonic osteotomy blades.

Number: 8

Where exactly? At the tip of the cutting blade? Could also be indicated with an arrow in Fig. 1 d.

Number: 9

How large was the set frequency increment?

Number: 10

Will this fit through a trocar?

Fig. 1 Design of the miniature ultrasonic surgical device and characterisation of the electromechanical responses. **(a)** Exploded view of the CAD model showing the components of the device. **(b)** front and side views of the fabricated device. **(c)** FEA predicted mode and **(d)** LDV measured mode, with red to blue indicating high to low displacement. **(e)** Impedance and phase characteristics as a function of frequency. **(f)** Harmonic response from LDV displacement measured at the tip of the blade.

amplitude of $46 \mu\text{m}$ peak-to-peak is measured at the tip of the blade, which is known to be compatible with effective bone cutting.

The characterization results of the miniature ultrasonic surgical device demonstrate its ability to achieve large vibration displacement amplitude of the blade tip, $40 \mu\text{m}$ being sufficient for bone cutting, while maintaining a stable connection to the surgical robot. The good agreement between the FEA model and the experimental measurements for the mode shape validates the design, with minimal unwanted vibrations and no bending in the PZT stack. Although the device exhibits a higher impedance magnitude and lower coupling coefficient compared to a BLT, indicating it will require an impedance matching circuit for resonance tracking, the measured mechanical quality

factor Q and achievable displacement amplitude at the blade tip indicate that it is a viable alternative configuration for an ultrasonic surgical device.

Electrical impedance matching circuit

Due to the high impedance magnitude (1927Ω) and strongly capacitive (-40.8°) characteristic of the surgical device at the resonance frequency, an impedance matching circuit is needed to maximise energy transfer to the cutting blade. Fig. 2 shows a simplified equivalent circuit of the surgical device at resonance. The device is represented by a Butterworth-Van Dyke (BVD) model [41], which is the most commonly used to represent an ultrasonic transducer at series resonance. Applying the BVD model to the ultrasonic surgical device, R_T represents the radiation and mechanical losses, L_T and C_T are the motion

 Number: 1

A reference should be added for this statement.

 Number: 2

Was operation at the anti- (parallel) resonant frequency considered? What are the arguments for/against this? There are commercial ultrasonic osteotomy systems that use fa drive strategy.

Fig. 2 Schematic showing the implementation of an LC impedance matching circuit between the electrical source and the miniature ultrasonic surgical device [1]

inductance and capacitance, respectively, and C_0 is the device's clamped capacitance. The device is excited by an electrical source through an LC impedance matching circuit. V_E represents the electrical source, R_E is the internal resistance of the electrical source, and L_m and C_m are the inductance and capacitance of the matching circuit, respectively.

It is known that losses in an ultrasonic surgical device are excitation-level dependent [38], [42], therefore applying optimal impedance matching parameters L_m and C_m derived from measurements at a low [3] static excitation level is unlikely to result in the largest achievable displacement amplitude when the device is operated at the high excitation level required to cut bone. To explore this, a parametric study is carried out at both low-level (or static) excitation, using an impedance analyser, and high-level excitation using a [4] resonance tracking system. The resonance tracker is a control system that continuously adjusts the driving frequency of the transducer to maintain resonance, under the continuously changing loading condition the device experiences in operation. This includes heating and cutting force that result in changes to the resonance frequency.

Impedance matching at low and high excitation levels

Fig. 3(a), (b) shows the effect of the two excitation levels on the electromechanical characteristics of the surgical device with an LC matching circuit implemented. Low-level (static) excitation measurements use an impedance analyser at 1 V, and high-level (dynamic) excitation measurements use a resonance tracking system at 30 V. The ranges of inductance, L_m , and capacitance, C_m , of the LC matching circuit are chosen to be 0 to 30 mH (with an increment of 2 mH) and 0 to 300 pF (with an increment of 30 pF), respectively. The impedance magnitude and phase, coupling coefficient k_{eff} , mechanical quality factor Q and displacement amplitude at the tip of the blade are calculated for each inductance L_m and capacitance C_m .

For static excitation matching, Fig. 3(a), the impedance magnitude drops significantly, from 1.9 k Ω to <100 Ω , as inductance increases, whereas it is unaffected by capacitance. A similar trend is observed for the phase, which changes from a capacitive characteristic at low inductance to a resistive characteristic at the highest inductance. Again, capacitance has little effect. The electromechanical coupling coefficient, k_{eff} , increases from [2] 0.05 to almost 0.3, peaking at the highest inductance and capacitance. This is mainly due to a large shift downwards of the resonance frequency as L_m and C_m increase, whereas the anti-resonance frequency is hardly changed. The mechanical Q exhibits its highest value (around 900) when L_m is around 20 mH; in this regime device losses will be minimised.

For dynamic excitation matching, 30 V excitation is selected, which is significantly higher than for the static excitation matching but also ensures the surgical device is excited in a linear regime. From Fig. 3(b), the impedance magnitude at steady-state vibration of the surgical device is around 4 k Ω when $L_m = 0$ and maintains this value until $C_m > 240$ pF when it increases up to 6.5 k Ω . In general, the impedance magnitude decreases at a high rate as L_m increases, whereas there is little effect of C_m . From measurements of the blade tip displacement, a peak amplitude of around 30 μm peak-to-peak is excited when the matching inductance is close to 20 mH, and it rapidly decreases either side of 20 mH. The maximum displacement is identified when $L_m = 20$ mH and $C_m = 270$ pF. This set of matching parameters is therefore adopted for the matching circuit to investigate the relationship between impedance magnitude, displacement amplitude and excitation voltage.

The results are shown in Fig. 3(c). The impedance magnitude grows quadratically from 250 Ω to 420 Ω , as the applied voltage is increased from 10 to 90 V in increments of 10 V. The optimal output impedance magnitude of the resonance tracking system is 400 Ω , meaning that the maximum energy transfer will be achieved if the impedance magnitude of the miniature surgical

Number: 1
add a point "."

Number: 2
It is difficult to understand why the highest Keff value "a measure of the surgical device's conversion efficiency from electrical energy to mechanical vibration" results in the lowest Q value and the smallest displacement (Fig. 3 b). Is there an explanation for this?

Number: 3
The word "static" can be misleading here. Does it refer to operation at DC voltage or rather to a very low deflection due to low power? I suggest omitting "static" and only distinguishing between low and high power.

Number: 4
You might consider adding the control (resonance tracking) to Fig. 2.

Fig. 3 Parametric study of the matching inductance and capacitance on the electromechanical responses of the miniature ultrasonic surgical device. **(a)** Impedance magnitude, phase, electromechanical coupling coefficient k_{eff} , and mechanical Q dependence on the matching inductance L_m and capacitance C_m , measured by an impedance analyser at 1 V excitation. **(b)** Impedance magnitude and displacement amplitude measured at the tip of the blade dependence on the matching inductance L_m and capacitance C_m , excited at 30 V with a resonance tracking system. **(c)** Impedance magnitude and displacement amplitude at the tip of the blade of the surgical device for increasing voltage from 10 to 90 V, adopting optimal impedance matching parameters $L_m = 20$ mH and $C_m = 270$ pF.

Fig. 4 Test rig for ultrasonic tissue cutting experiments. (a) Schematic showing test rig components. (b) Image of the test rig. (c) Close-up view of the linear actuator and prototype bone surgery device showing the directions of the cutting forces, F_x and F_z , cutting speed, V_L , stroke length, L , penetration rate, V_z , and target penetration depth, D .

device driven continuously for bone cutting were close to this value. A linear increase in the displacement amplitude of the blade is observed, reaching $60 \mu\text{m}$ peak-to-peak at 90 V . However, it should be noted that this is not a limit of the ultrasonic surgical device, but rather a limitation of the LDV measurement set-up.

Even though the optimal set of impedance matching parameters has been identified ($L_m = 20 \text{ mH}$ and $C_m = 270 \text{ pF}$) from the dynamic excitation matching, the impedance magnitude of the device has also increased significantly (from 250 to 420Ω), highlighting the influence of excitation level on impedance. Additionally, device loading during bone cutting will likely result in a further increase in impedance, leading to a reduction in efficiency. Therefore, to understand how the resonance tracking and impedance matching circuit perform during bone cutting, experiments were focused on the effects of cutting speed, penetration rate, stroke, and total cutting depth.

Ultrasonic bone cutting experiments

Prior to integration with the surgical robot, a dedicated test rig was developed to perform controlled *in vitro* cutting tests on

bone samples, as shown in detail in Fig. 4. The test rig is configured to measure the following parameters: lateral cutting force and bone penetration force (measured using a force sensor, Kistler), impedance magnitude and consumed power (measured by the resonance tracking system Piezodrive, PDUS210-800), and temperature at both the cut site and the piezoelectric stack measured using a thermal imaging camera TIM 160, Micro-Epsilon). Fresh porcine rib was used for the tissue cutting experiments, because of its similarity in density, porosity, microstructure, and composition to human bones [43], [44], [45]. For all tests, the cutting speed, V_L , is set at 3 mm/s , which guarantees cutting stability and control within the accuracy of the test rig. The stroke, L , is 2 mm and target depth, D , is 2 mm , selected as commensurate with the size of the dimensions of the bone sample. The vertical engagement speed, which is the device penetration rate, V_z , is set to 5 , 10 and $15 \mu\text{m/s}$, as these maintain sufficiently low cutting force in the penetration direction to be able to maintain control of the test rig stability. The blade tip displacements are set to 0 , 30 and $60 \mu\text{m}$ peak-to-peak.

Number: 1

This argument is not entirely comprehensible, as the both the MSA100 and OFV 303 can measure up to 10 m/s, which at a frequency of 23896 Hz would correspond to a peak-to-peak displacement of 133 μm .

Number: 2

Has the emissivity been calibrated? The cutting side (bone or cutting blade) may have a different emissivity than the PZT stack.

Number: 3

The description of the setup and the cutting process is clear and comprehensible. However, it is not entirely clear why this type of sawing motion was chosen for the test, as this is rarely the case when using ultrasonic osteotomy devices. Could a more detailed explanation be provided in this regard? Furthermore, such a sawing motion and penetration rate are difficult to implement in robot-assisted surgery and are likely to be too slow and imprecise for clinical requirements.

Fig. 5 Ultrasonic tissue cutting results on the test rig. **(a)** Forces F_x and F_z at displacement amplitudes 0, 30 and 60 μ m and penetration rates 5, 10 and 15 μ m/s. **(b)** Impedance magnitude, power consumption, and temperature measured at the cut site at amplitudes of 30 and 60 μ m and penetration rates 5, 10 and 15 μ m/s.

Fig. 6 Microscopic images of the dissected porcine bone samples and depths of cut at three displacement amplitudes 0, 30 and 60 μm and at three penetration rates 5, 10 and 15 $\mu\text{m/s}$.

The results are shown in Fig. 5. It can be observed that for the larger amplitude of 60 μm a significant reduction in both the lateral cutting force F_x and the vertical penetration force F_z are achieved compared to the non-ultrasonic cutting. This reduction is particularly noticeable at the slowest penetration rate, $V_z = 5 \mu\text{m/s}$, where the maximum force remains below 1 N, highlighting the advantage of using a higher blade displacement amplitude.

At the higher penetration rate, the significant force reduction is lost, however 60 μm amplitude still achieves a 50% reduction in both force components to non-ultrasonic cutting, which stay under 4 N. This suggests that at this penetration rate, the bone removal efficiency decreases, especially when the blade engages deeper with the bone. Furthermore, 30 μm amplitude exhibits similar or **even larger forces than the non-ultrasonic forces**, indicating that lower amplitude at this penetration rate provides no significant advantage.

As the penetration rate increases to 15 $\mu\text{m/s}$, the advantage of ultrasonic cutting is completely lost. At 60 μm amplitude, the sudden rise in F_x at 65 seconds suggests a temporary loss of resonance tracking, followed by oscillatory characteristics, indicating the system attempts to restore the amplitude as the blade is engaged deeper with the bone. This leads F_z to rise sharply due to increased cutting depth. Unexpectedly, 30 μm amplitude results in lower forces than both the 60 μm amplitude and the non-ultrasonic cutting, with values even lower than those observed at penetration rates of 5 and 10 $\mu\text{m/s}$. **This may be attributed to cutting occurring in a region of the cortical bone where mechanical properties are more favourable, which identifies an issue in tests inconsistency of bone properties.**

The impedance magnitude for 30 μm amplitude starts approximately 100 Ω lower than for 60 μm . Additionally, the starting power difference is about 2 W, highlighting the influence of the excitation level. For all engagement rates, both displacement amplitudes result in a sudden increase in impedance magnitude and decrease in power (more pronounced for 60 μm than 30 μm). These changes occur at around 200, 75,

and 40 seconds for 5, 10, and 15 $\mu\text{m/s}$ engagement rates, respectively. This is due to full engagement of all the serrated teeth of the blade with the bone, increasing the surface contact area. Under these conditions, the **resonance tracking system** was unable to maintain stability.

Unlike the 5 $\mu\text{m/s}$ penetration rate, the other two penetration rates exhibit noticeable oscillatory behaviour at a frequency much lower than the cutting speed, as the blade penetrates deeper into bone. These oscillations start at 110 seconds for the 10 $\mu\text{m/s}$ penetration rate and at 65 seconds for the 15 $\mu\text{m/s}$ penetration rate. This clearly indicates a temporary loss of resonance, with the tracking system attempting to recover it. The thermal response further supports these findings; for 60 μm and 5 $\mu\text{m/s}$ the temperature steadily increases as the blade reaches full depth, indicating a strong vibro-impact micro-sawing interaction between the blade and bone [46], [47], [48]. In contrast, for both 10 and 15 $\mu\text{m/s}$ penetration rates, temperature decreases occur at times corresponding to the changes in impedance and power, reflecting a reduced vibro-impact micro-sawing interaction between the blade and bone.

Fig. 6 shows the microscopic images of the bone cuts made at 5 $\mu\text{m/s}$ penetration rate with increasing blade amplitude, while the table summarizes cutting depths for all tested parameters. For this test-rig set-up, it is optimal to operate the device at a high amplitude and slow penetration rate. This approach generates extremely low forces and stable electrical responses, evidencing a strong vibro-impact micro-sawing interaction between the blade and bone. The high temperature also indicates strong blade/bone interaction and is **generally mitigated in ultrasonic bone surgery devices by incorporating cooling.**

Device integration with a Kuka surgical robot

From the findings of tests in the test rig, 60 μm blade amplitude and 5 $\mu\text{m/s}$ penetration rate are selected for testing the miniature device on the Kuka surgical robot. The set-up for *in vitro* robotically assisted bone cutting experiments is shown in Fig. 7. Cutting experiments are performed with cutting speed, V_L ,

Number: 1

Can an outlook be provided as to whether the resonance tracking system, the transducer prototype or the cutting blade requires optimization?

Number: 2

Is there an explanation for the fact that at 10 $\mu\text{m/s}$, the forces at 30 μm are even higher than with the non-activated transducer, and then at 15 $\mu\text{m/s}$ they are lower compared to 0 μm and 60 μm or lower cutting speeds?

One possible reason, as also mentioned later in this paragraph, could be that the material tested is biological samples, which by nature exhibit a high variation in mechanical properties (e.g., thickness of cortical bone). There are also synthetic bone models that may allow for more standardized testing of device performance.

See for example: O. F. Okaya, J. Burger, and M. Hofmann, "Additively Manufactured Ultrasonic Osteotomy Inserts for Improved Temperature Control and Tissue Preservation," 2024 52nd Annual Ultrasonic Industry Association Symposium (UIA), Dublin, Ireland, 2024, pp. 1-5, doi: 10.23919/UIA60812.2024.10716027.

Number: 3

Were the measurements repeated? If so, what is the sample size and what is the standard deviation? If not, further measurements on biological or synthetic samples may be necessary in order to make a proper assessment.

Number: 4

Why was an irrigation system, which is typically used for flushing and cooling during ultrasonic osteotomy, not implemented in the test bench?

Fig. 7 Integration of the miniature ultrasonic surgical device with Kuka surgical robot for bone cutting tests. **(a)** Schematic of the robotic cutting platform. **(b)** Image of the experimental set-up. **(c)** Close-up view of the device integrated with robot showing directions of the cutting forces, F_x , F_y and F_z , cutting speed, V_L , stroke length, L , cut depth increment Δ_z , and target penetration depth D .

set at 2, 3, and 4 mm/s, depth increment, Δ_z , at 20 and 40 μm per cycle, and target depth, D , at 1 and 1.5 mm. The results are presented in Fig. 8.

All three measured force components, F_x , F_y , and F_z , remain below 1 N, despite increases in cutting speed, depth increment per cycle, and target depth. The lateral force F_y is insignificant (close to 0 N) across all cutting parameters, suggesting that the increased freedom and compliance of the motion-driving mechanism (Kuka surgical robot) allow the miniature device to create a cut slightly wider than the blade itself, minimizing friction with the bone. This could also mean that no significant bending motion occurs in the blade.

At higher cutting speeds (3 and 4 mm/s), the vertical penetration force, F_z , is less than 0.2 N. This suggests that the combined vibro-impact and micro-sawing motion of the blade, superimposed onto the reciprocal sawing speed, enhance the dynamic interaction with the bone, promoting fracture and increasing bone material removal rate. Meanwhile, the horizontal cutting force, F_x , steadily rises as the blade engages

deeper with the bone, reaching approximately 0.5 N. This indicates that the serrated teeth of the blade are effectively removing bone material with each incremental depth increase per cycle.

The impedance magnitude starts at 370 Ω and increases significantly as the device operates at higher speeds and larger incremental depths per cycle. This increase leads to a drop in the acoustic power from an initial 4 W to 3 W. A more pronounced change in impedance magnitude is observed at 250 seconds, for a cutting speed 3 mm/s, in incremental depth 20 μm , and target depth 1.5 mm. Under these conditions, power fluctuations occur, temporarily dropping to nearly 2 W. This variation is likely to occur when the blade encounters a harder bone region, due to the material properties of bone not being constant over the cut site.

Temperature at the PZT stack gradually increases to approximately 60°C for all cutting parameters. At the cutting site, the highest temperature is recorded for the highest cutting speed, which generates higher friction.

Fig. 8 Ultrasonic tissue cutting results facilitated with Kuka surgical robot: (a) Force F_x , F_y and F_z . (b) Change in impedance magnitude and power consumption. (c) Temperature measured at the piezoelectric stack and the cutting site at three cutting speeds 2, 3 and 4 mm/s and two depth increments per cycle 20 and 40 μm .

Fig. 9 Image and measurement of cuts in porcine bone samples at three cutting speeds, 2, 3 and 4 mm/s and two depth increments per cycle, 20 and 40 μm .

Microscopic images and bone cut profiles are shown in Fig. 9. The cut edges remain intact, with no visible burrs or microdamage for all three parameter sets. Additionally, the blade has nearly fully penetrated the cortical layer at the 1.5 mm target depth, as indicated by the darker colouration at the bottom of the cut, which signifies marrow exposure. The measured cut depths are 0.78, 1.21, and 1.42 mm, while the corresponding widths are 0.56, 0.53, and 0.54 mm, for all three parameter sets.

Results from the *in vitro* bone cutting experiments with the Kuka surgical robot differ from the observations made using the test rig. This highlights the advantages of the high precision and additional degrees-of-freedom, which enables compliant interactions between the blade and bone. This setup provides more controlled engagement accuracy, which is difficult to achieve using the rigid test rig.

A higher cutting speed and greater incremental depth per cycle enhance the interaction between the blade and bone, leading to increased temperature at the cutting site. This highlights the need for cooling, which is typically incorporated in ultrasonic surgical devices. **Future research will focus on the optimization of blade geometries, tailored to specific surgical procedures. Additionally, integrating haptic mechanisms with force feedback strategies could improve the motion control of the surgical robot, enabling more precise bone cutting along pre-defined pathways.**

Methods

Fabrication of the surgical device

Four hard piezoelectric PZT rings (PIC-181, PI Ceramic) of dimension outer diameter 10 mm, inner diameter 5 mm and thickness 2 mm are used, with material properties defined in the literature [4]. Four copper electrodes, one metal frame made from titanium grade 5 alloy Ti-6Al4V, one threaded bar and two nuts made from A4 tool steel, are all thoroughly cleaned using isopropyl alcohol (IPA) solution prior to the fabrication process. The PZT ring stack is then assembled and inserted into the

metal frame, before the threaded bar and two nuts are installed (see Fig. 1 (a)). Two wires are attached to the live and ground terminals of the PZT stack. Torque is gradually applied to the nuts and threaded bar, increasing from 0.5 to 3.0 Nm in increments of 0.5 Nm, to achieve an ultimate pre-stress of around 30 MPa [49]. Impedance is recorded for each applied torque increment during fabrication, with a stabilisation of resonance frequency and impedance magnitude being an indicator of sufficient pre-stress. The device is then allowed to settle for a few weeks, allowing for the electrical characteristics to reach a steady-state, confirmed by no further change in impedance.

Electromechanical characteristics analysis

The impedance characteristics of the miniature ultrasonic surgical device are measured using an impedance analyser (4294A, Agilent) with 1 V swept signal applied across the bandwidth of the resonance frequency of the device.

Experimental modal analysis of the surgical device is conducted using an MSA-100 3-D laser Doppler vibrometer (Polytec), and the results are compared with the vibration mode shapes predicted in finite element analysis (FEA) using Abaqus-Simulia (Dassault Systèmes) software.

Harmonic analysis experiments are performed to understand how the ultrasonic displacement amplitude at the tip of the blade varies with excitation level. The surgical device is excited with a frequency sweep from below to above the resonance frequency. A burst sine wave is used, generated by a signal generator (Agilent 33210A) and amplified by a power amplifier (HFVA-62). The vibration displacement response at the tip of the blade is measured using a 1-D laser Doppler vibrometer (OFV 303, Polytec).

Design of the electrical impedance matching circuit

The impedance analyser (4294A, Agilent) which has an output impedance of 50 Ω was used for the low level (static) excitation analysis. For high level (dynamic) excitation, a resonance tracking system (PDUS210-800, Piezodrive) is employed with an optimal output impedance magnitude 400 Ω . For both

 Number: 1
", USA"? To keep the naming consistent with the "Results" section. In general, the manuscript should be checked for consistency in

 naming. Number: 2
The article is about a miniature ultrasonic surgical device based on a flexensional configuration with a pre-stressed PZT stack (as per the title). In addition, the article specifically describes two test modalities with two different test setups. It is a valid approach to test the feasibility and functionality of a surgical instrument in the target application (bone cutting). However, it would be beneficial to also have a commercially available device or a traditional Langevin transducer as a comparison in the same test environment to clearly demonstrate the added value of the novel concept. Although it is generally demonstrated that the present concept has potential, it is not entirely clear which advantages can be attributed to the novel transducer concept, the impedance matching and the corresponding resonance tracking, or the shape of the cutting blade. I therefore recommend supplementing the discussion of these points by taking relevant literature into account.

excitation regimes, an inductance decade box (DL07, ELC) and a capacitance decade box (DC05, ELC) are used for the parametric study.

Bone cutting test rig

Fig. 4 shows the ultrasonic tissue cutting test rig. The surgical device is fixed to a force sensor (9311b, Kistler), which is attached horizontally to an L-shape plate, which is then connected to another force sensor (9321b, Kistler) via an L-shape plate that is deployed in the vertical direction (see Fig. 4 (c)). The assembly is then fixed to the crossbeams which are driven by a stepper motor controlled linear actuator (GLA750-STEP-20-3-285-390, stroke length 285 mm, Gimson Robotics), which is mounted between the table and the upper fixed crossbeams. Device penetration rate in the vertical direction, denoted as V_z , is facilitated with a bespoke motion control circuit, and the target depth D can also be pre-defined. The crossbeams move in a purely vertical direction with the support of four linear rails and four embedded needle bearings to minimize lateral motions of the surgical device.

A fresh porcine rib bone is held firmly in a vice, as shown in Fig. 4 (c), which was mounted to another DC driven linear actuator (GLA750-P, stroke length 100 mm, Gimson Robotics) that drives the bone in a sawing action at a speed V_L and with a stroke length L in the horizontal direction. The cutting forces in both horizontal direction, F_x , and vertical direction, F_z , are measured by the two force sensors simultaneously. Additionally, a thermal camera (TIM 160, Micro-Epsilon) is mounted close to the cutting site, focusing on the tip of the blade to record the cutting temperature. The bone cuts are measured by a microscope (AmScope).

Robotic bone cutting platform

The experimental arrangement of the integration of the miniature ultrasonic surgical device with a Kuka surgical robot is presented in Fig. 7.

The ultrasonic surgical device is attached to a six-axis loadcell (Nano17 SI-25-0.25, ATI Industrial Automation) via an aluminium fixture to measure the forces in the cutting direction (F_x), the transverse direction (F_y), and the engagement direction (F_z). The device and loadcell are fixed to a **6x degree-of-freedom positioning hexapod with 0.1 μ m precision (Solano, Symétrie), which is then connected to the flange of a Kuka surgical robot (LBR iiwa 14 R820, Kuka AG), capable of seven axes of motion.** The entire assembly is mounted to a bench. A vice is fixed to a metal extrude that is bolted vertically to the bench, which is used to clamp a fresh porcine rib bone. A thermal camera (TIM 640, Micro-Epsilon) and a high-resolution video camera (a2A1920-160 μ cBAS, ace 2 R, Basler) are mounted to another metal extrude to record the cutting temperature and video each test. The dissected bone sample is analyzed using a digital microscope (VHX-6000, Keyence).

During operation, the surgical device, loadcell and hexapod assembly is deployed by the robot from a 'home' position (at some distance from the cutting site) to the 'ready-to-cut' position (with the middle tooth of the blade serrations aligned and in contact with the bone). Thereafter, the surgical device is

powered, and the high precision hexapod is activated to perform bone cutting. The sequence of cutting is that once the surgical device is deployed at the 'ready-to-cut' position, the robot will drop 500 μ m to ensure the device is in contact with the bone, which is associated with a small increase in the force F_z . The linear motion of the hexapod is activated, and the surgical device is driven forward at speed V_L for half of the stroke length L and backward for the full stroke length, followed by forward for other half, returning to the initial position. Next, the hexapod lowers the surgical device by one depth increment Δ_z , and the cutting cycle repeats until the target depth D is reached.

Bone sample preparation

Fresh porcine ribs are procured and excarnated using a scalpel, and were stored in isotonic Phosphate Buffered Saline to maintain hydration. The bone sample is used in tests on the same day as procurement.

Acknowledgement

This work was supported by an EPSRC Programme Grant, Ultrasurge – Surgery enabled by ultrasonics, EP/R045291/1.

References

- [1] P. Hennet, "Piezoelectric bone surgery: A review of the literature and potential applications in veterinary oromaxillofacial surgery," *Frontiers in Veterinary Science*, vol. 2, no. 8, pp. 1–7, 2015.
- [2] B. Anderson, K. Mozaffari, C. H. Foster, A. A. Jaco, and M. K. Rosner, "The ultrasonic bone scalpel does not outperform the high-speed drill: A single academic experience," *World Neurosurgery*, vol. 185, pp. e387–e396, 2024.
- [3] G. J. Ledderose, N. Thon, W. Rachinger, and C. S. Betz, "Use of an ultrasonic aspirator in transnasal surgery of tumorous lesions of the anterior skull base," *Interdisciplinary Neurosurgery*, vol. 18, pp. 1–4, 2019.
- [4] X. Li, T. Stritch, K. Manley, and M. Lucas, "Limits and opportunities for miniaturizing ultrasonic surgical devices based on a Langevin transducer," *IEEE Transactions on Ultrasonics, Ferroelectrics, and Frequency Control*, vol. 68, no. 7, pp. 2543–2553, 2021.
- [5] G. L. Magrin, E. A. Sigua-Rodriguez, D. R. Goulart, and L. Asprino, "Piezosurgery in bone augmentation procedures previous to dental implant surgery: A review of the literature," *The Open Dentistry Journal*, vol. 9, pp. 426–430, 2015.
- [6] M. Robiony, F. Polini, F. Costa, N. Zerman, and M. Politi, "Ultrasonic bone cutting for surgically assisted rapid maxillary expansion (SARME) under local anaesthesia," *International Journal of Oral & Maxillofacial Surgery*, vol. 36, no. 3, pp. 267–269, 2007.
- [7] Y. Kato, N. Saito, K. Niimi, D. Saito, H. Sakuma, D. Hasebe, W. Katagiri, and T. Kobayashi, "A comparison and evaluation of the use of ultrasonic cutting devices with conventional powered instruments in orthognathic surgery," *Advances in Oral and Maxillofacial Surgery*, vol. 2, pp. 1–5, 2021.
- [8] M. M. Rastelli Jr, C. D. Pinheiro-Neto, J. C. Fernandez-Miranda, E. W. Wang, C. H. Snyderman, and P. A. Gardner, "Application of ultrasonic bone curette in endoscopic endonasal skull base surgery: technical note," *Journal of Neurological Surgery–Part B*, vol. 75, no. B2, pp. 90–95, 2014.

 Number: 1

A more detailed explanation is needed as to why a hexapod is used for the cutting movement and why not just the Kuka robot.

- [9] R. Al-Mahfoudh, E. Qattan, J. R. Ellenbogen, M. Wilby, C. Barrett, and T. Pigott, “Applications of the ultrasonic bone cutter in spinal surgery – our preliminary experience,” *British Journal of Neurosurgery*, vol. 28, no. 1, pp. 56–60, 2014.
- [10] K. R. Renjith, N. K. Eamani, D. C. Raja, and A. P. Shetty, “Ultrasonic bone scalpel in spine surgery,” *Journal of Orthopaedics*, vol. 41, pp. 1–7, 2023.
- [11] X. Hu, D. D. Ohnmeiss, and I. H. Lieberman, “Use of an ultrasonic osteotome device in spine surgery: experience from the first 128 patients,” *European Spine Journal*, vol. 22, pp. 2845–2849, 2013.
- [12] G. Dagnino and D. Kundrat, “Robot-assistive minimally invasive surgery: trends and future directions,” *International Journal of Intelligent Robotics and Applications*, vol. 8, pp. 812–826, 2024.
- [13] N. Simaan, R. M. Yasin, and L. Wang, “Medical technologies and challenges of robot-assisted minimally invasive intervention and diagnostics,” *Annual Review of Control, Robotics, and Autonomous Systems*, vol. 1, pp. 465–490, 2018.
- [14] A. L. Orekhov, C. Abah, and N. Simaan, “Snake-like robots for minimally invasive, single-port, and intraluminal surgeries,” *The Encyclopedia of Medical Robotics*, vol. 1, chapter. 8, pp. 203–243, 2018.
- [15] M. Runciman, A. Darzi, and G. P. Mylonas, “Soft robotics in minimally invasive surgery,” *Soft Robotics*, vol. 6, no. 4, pp. 423–443, 2019.
- [16] J. H. Palep, “Robotic assisted minimally invasive surgery,” *Journal of Minimal Access Surgery*, vol. 5, no. 1, pp. 1–7, 2009.
- [17] O. T. Okusanya, I. S. Sarkaria, N. R. Hess, K. S. Nason, M. V. Sanchez, R. M. Levy, A. Pennathur, and J. D. Luketich, “Robotic assisted minimally invasive esophagectomy (RAMIE): the University of Pittsburgh Medical Center initial experience,” *Annals of Cardiothoracic Surgery*, vol. 6, no. 2, pp. 179–185, 2017.
- [18] A. Brodie and N. Vasdev, “The future of robotic surgery,” *Robotics: Annals of the Royal College of Surgeons of England*, vol. 100, pp. 4–13, 2018.
- [19] V. Vitiello, K. W. Kwok, and G. Z. Yang, “Introduction to robot-assisted minimally invasive surgery (MIS),” *Medical Robotics*, pp. 1–40, 2012.
- [20] F. J. Kim, M. F. Chammas Jr, E. Gewehr, M. Morihisa, F. Caldas, E. Hayacibara, M. Baptistussi, F. Meyer, and A. C. Martins, “Temperature safety profile of laparoscopic devices: Harmonic ACE (ACE), Ligasure V (LV), and plasma trisector (PT),” *Surgical Endoscopy and Other Interventional Techniques*, vol. 22, pp. 1464–1469, 2008.
- [21] C. Koch, T. Friedrich, F. Metternich, A. Tannapfel, H. P. Reimann, and U. Eichfeld, “Determination of temperature elevation in tissue during the application of the harmonic scalpel,” *Ultrasound in Medicine & Biology*, vol. 29, no. 2, pp. 301–309, 2003.
- [22] A. B. Kassam, J. A. Engh, A. H. Mintz, and D. M. Prevedello, “Completely endoscopic resection of intraparenchymal brain tumors,” *Journal of Neurosurgery*, vol. 110, no. 1, pp. 116–123, 2009.
- [23] H. J. Kim, G. S. Choi, J. S. Park, and S. Y. Park, “Comparison of surgical skills in laparoscopic and robotic tasks between experienced surgeons and novices in laparoscopic surgery: an experimental study,” *Annals of Coloproctology*, vol. 30, no. 2, pp. 71–76, 2014.
- [24] T. B. Limperg, V. Y. Novoa, H. L. Curlin, and S. Veersema, “Laparoscopic trocars: marketed versus true dimensions – A descriptive study,” *Journal of Minimally Invasive Gynecology*, vol. 31, no. 4, pp. 304–308, 2024.
- [25] S. Delibegovic, “Minimising in minimally invasive surgery through the use of a novel and flexible super elastic titanium needle suitable for a 3.5- and 5-mm trocar,” *Journal of Minimal Access Surgery*, vol. 18, no. 1, pp. 161–163, 2022.
- [26] C. H. Sherman and J. L. Butler, “Transducers and arrays for underwater sound,” *Springer*, 2007.
- [27] F. Bejarano, M. Lucas, R. Wallace, A. M. Spadaccino, and H. Simpson, “Ultrasonic cutting device for bone surgery based on a cymbal transducer,” *Physics Procedia*, vol. 63, pp. 120–126, 2015.
- [28] F. Bejarano, A. Feeney, R. Wallace, H. Simpson, and M. Lucas, “An ultrasonic orthopaedic surgical device based on a cymbal transducer,” *Ultrasonics*, vol. 72, pp. 24–33, 2016.
- [29] F. Bejarano, A. Feeney, and M. Lucas, “A cymbal transducer for power ultrasonics applications,” *Sensors and Actuators A: Physical*, vol. 210, pp. 182–189, 2014.
- [30] J. Zhang, W. J. Hughes, P. Bouchilloux, R. J. Meyer Jr, K. Uchino, and R. E. Newnham, “A class V flextensional transducer: the cymbal,” *Ultrasonics*, vol. 37, no. 6, pp. 387–393, 1999.
- [31] T. Hemsell, E. G. Lierke, W. Littmann, and T. Morita, “Various aspects of the placement of a piezoelectric material in composite actuators, motors, and transducers,” *Journal of the Korean Physical Society*, vol. 57, no. 4, pp. 933–937, 2010.
- [32] A. Mathieson, A. Cardoni, N. Cerisola, and M. Lucas, “The influence of piezoceramic stack location on nonlinear behavior of Langevin transducers,” *IEEE Transactions on Ultrasonics, Ferroelectrics, and Frequency Control*, vol. 60, no. 6, pp. 1126–1133, 2013.
- [33] V. K. Astashev, K. A. Pichugin, X. Li, A. Meadows, and V. I. Babitsky, “Resonant tuning of Langevin transducers for ultrasonically assisted machining applications,” *IEEE Transactions on Ultrasonics, Ferroelectrics, and Frequency Control*, vol. 67, no. 9, pp. 1888–1896, 2020.
- [34] A. Caronti, R. Carotenuto, and M. Pappalardo, “Electromechanical coupling factor of capacitive micromachined ultrasonic transducers,” *The Journal of the Acoustical Society of America*, vol. 113, pp. 279–288, 2003.
- [35] J. Kim and J. Lee, “Parametric study of bolt clamping effect on resonance characteristics of Langevin transducers with lumped circuit models,” *Sensors*, vol. 20, no. 7, pp. 1–9, 2020.
- [36] S. Lin, “Study on the multifrequency Langevin ultrasonic transducer,” *Ultrasonics*, vol. 33, no. 6, pp. 445–448, 1995.
- [37] A. Pérez-Sánchez, J. A. Segura, C. Rubio-Gonzalez, L. A. Baldenegro-Pérez, and J. A. Soto-Cajiga, “Numerical design and analysis of a Langevin power ultrasonic transducer for acoustic cavitation generation,” *Sensors and Actuators A: Physical*, vol. 311, pp. 1–12, 2020.
- [38] M. Tsuchida and T. Morita, “Non-contact mechanical Q-factor measurement system based on electromagnetic acoustic transducer,” *Precision Engineering*, vol. 91, pp. 390–395, 2024.
- [39] J. Lee and J. Kim, “Theoretical and empirical verification of electrical impedance matching method for high-power transducers,” *Electronics*, vol. 11, no. 2, pp. 1–13, 2022.
- [40] A. Mathieson, A. Cardoni, N. Cerisola, and M. Lucas, “Understanding nonlinear vibration behaviours in high-power ultrasonic surgical devices,” *Proceedings of the Royal Society A: Mathematical, Physical and Engineering Sciences*, vol. 471, no. 2176, pp. 1–19, 2015.

- [41] M. Garcia-Rodriguez, J. Garcia-Alvarez, Y. Yañez, M. J. Garcia-Hernandez, J. Salazar, A. Turo, and J. A. Chavez, “Low cost matching network for ultrasonic transducers,” *Physics Procedia*, vol. 3, no. 1, pp. 1025–1031, 2010.
- [42] X. Li, N. G. Fenu, N. Giles-Donovan, S. Cochran, and M. Lucas, “Can Mn:PIN-PMN-PT piezocrystal replace hard piezoceramic in power ultrasonic devices?,” *Ultrasonics*, vol. 138, pp. 1–16, 2024.
- [43] C. G. Finkemeier, “Bone-grafting and bone-graft substitutes,” *The Journal of Bone & Joint Surgery*, vol. 84, no. 3, pp. 454–464, 2002.
- [44] G. K. B. Sándor, T. C. Lindholm, and C. M. L. Clokie, “Bone regeneration of the cranio-maxillofacial and dento-alveolar skeletons in the framework of tissue engineering,” *Tissue Engineering*, Chapter. 7, pp. 1–46, 2003.
- [45] M. Figueiredo, J. Henriques, G. Martins, F. Guerra, F. Judas, and H. Figueiredo, “Physicochemical characterization of biomaterials commonly used in dentistry as bone substitutes - comparison with human bone,” *Journal of Biomedical Materials Research Part B: Applied Biomaterials*, vol. 92B, no. 2, pp. 409–419, 2010.
- [46] V. K. Astashev and V. I. Babitsky, and K. Khusnutdinova, “Ultrasonic processes and machines: dynamics, control and applications,” *Springer*, 2007.
- [47] V. I. Babitsky, “Theory of vibro-impact systems and applications,” *Springer*, 1998.
- [48] V. K. Astashev and V. I. Babitsky, “Ultrasonic cutting as a nonlinear (vibro-impact) process,” *Ultrasonics*, vol. 36, no. 1–5, pp. 89–96, 1998.
- [49] F. J. Arnold and S. S. Mühlen, “Mechanical pre-stressing in ultrasonic piezotransducers,” *Ultrasonics*, vol. 39, no. 1, pp. 7–11, 2001.